# ER-associated VAP27-1 and VAP27-3 proteins functionally link the lipid-binding ORP2A at the ER-chloroplast contact sites

Luciana Renna[1,2], Giovanni Stefano [1,3,4], Maria Paola Puggioni[1,5], Sang-Jin Kim [1,3,6], Anastasiya Lavell[1], John E. Froehlich[1,7], Graham Burkart[1], Stefano Mancuso[2,8], Christoph Benning [1,3,7] & Federica Brandizzi [1,3,6] ✉

The plant endoplasmic reticulum (ER) contacts heterotypic membranes at membrane contact sites (MCSs) through largely undefined mechanisms. For instance, despite the well-established and essential role of the plant ER-chloroplast interactions for lipid biosynthesis, and the reported existence of physical contacts between these organelles, almost nothing is known about the ER-chloroplast MCS identity. Here we show that the Arabidopsis ER membrane-associated VAP27 proteins and the lipid-binding protein ORP2A define a functional complex at the ER-chloroplast MCSs. Specifically, through in vivo and in vitro association assays, we found that VAP27 proteins interact with the outer envelope membrane (OEM) of chloroplasts, where they bind to ORP2A. Through lipidomic analyses, we established that VAP27 proteins and ORP2A directly interact with the chloroplast OEM mono-galactosyldiacylglycerol (MGDG), and we demonstrated that the loss of the VAP27-ORP2A complex is accompanied by subtle changes in the acyl composition of MGDG and PG. We also found that ORP2A interacts with phytos-terols and established that the loss of the VAP27-ORP2A complex alters sterol levels in chloroplasts. We propose that, by interacting directly with OEM lipids, the VAP27-ORP2A complex defines plant-unique MCSs that bridge ER and chloroplasts and are involved in chloroplast lipid homeostasis.

The endoplasmic reticulum (ER) is responsible for essential biological functions that include synthesis of one-third of the cellular proteome, priming the glycosylation of secretory proteins, and lipid synthesis[1,2]. The ER is shaped in tubules and cisternae and closely associates with other membranes via protein–protein and protein–lipid interactions[3,4]. Such interactions occur at ER membrane subdomains that are generally referred to as membrane contact sites (MCSs)[5,6]. In plant cells, the MCSs between the ER and mitochondria, and ER and the plasma membrane

(PM) are the most characterized to date[5,7–13]. Not much is known about the physical association of the ER with other organelles despite their reported existence in plant cells[14,15]. ER and chloroplasts associate at plastid-associated membranes (PLAMs) through yet-unknown mechanisms[16,17]. To date, hypothetical PLAMs have been visualized with artificial reporters based on reconstitution of complementary protein fusions targeted to the ER and chloroplasts[17], and with localization of a *Brassica napus* lipase, BnCLIP1, in transient expression in

[1]MSU-DOE Plant Research Lab, Michigan State University, East Lansing, MI, USA. [2]Department of Horticulture, University of Florence, Florence, Italy. [3]Department of Plant Biology, Michigan State University, East Lansing, MI, USA. [4]Department of Biology, University of Florence, Florence, Italy. [5]Department of Plant Physiology, Umeå Plant Science Centre, Umeå University, Umeå, Sweden. [6]Great Lakes Bioenergy Research Center, Michigan State University, East Lansing, MI, USA. [7]Biochemistry and Molecular Biology Department, Michigan State University, East Lansing, MI, USA. [8]Fondazione per il Futuro delle Città, Florence, Italy. ✉e-mail: fb@msu.edu

tobacco leaves[18]. Therefore, information on the nature of endogenous PLAMs is still missing. Experiments based on optical laser tweezers on isolated *Arabidopsis thaliana* chloroplasts from protoplasts expressing a fluorescent ER lumenal marker have shown that the chloroplast–ER association can withstand a pulling force exceeding 400 pN and that protease treatment lowers such force[16]. Therefore, ER–chloroplast physical connections are most likely mediated by interactions among proteins[16], whose identity is still unknown. Because PLAMs are structures unique to plastid-bearing cells, it is difficult to infer their composition directly from other species, such as metazoans and yeast.

Functionally, the ER and chloroplasts interact to synthesize all cellular lipids, and the interaction is particularly extensive for the lipids of the thylakoid membrane lipids. The thylakoid lipids include the glycoglycerolipids mono- and digalactosyl diacylglycerol (MGDG and DGDG, respectively), which are unique to the OEM, and the anionic lipid sulfoquinovosyl diacylglycerol (SQDG) and phospholipid phosphatidylglycerol (PG)[19]. Glycerolipid biosynthesis requires enzymes in the ER and chloroplasts. Initially, fatty acids are synthesized within the chloroplasts and are either directly incorporated into chloroplast lipids or exported to the ER for the synthesis of extraplastidic phospholipids and triacylglycerol. Precursors derived from ER phospholipids return to chloroplasts to be assembled into galactolipids[2,20]. While it has been hypothesized that PLAMs may be involved in ER–chloroplast bidirectional transport of acyl lipids[2,16], or of DGDGs, which substitute for phospholipids in the PM, mitochondria, and tonoplast in conditions of phosphate limitation[21–24], it is still to be experimentally established if PLAMs are involved in the inter-organellar transport of lipids, including sterols.

Ubiquitous constituents of the MCSs are proteins of the family of syntaxin-like vesicle-associated membrane protein (VAMP)-associated proteins (VAPs)[25,26], which are represented by the yeast Scs2, metazoan VAPs, and the plant VAP27 proteins[25]. These are proteins anchored to the ER membrane by a single transmembrane domain at the C-terminus and have a cytosolic major sperm (MSP) domain and a coiled-coil domain at the N-terminus[25,27]. The Arabidopsis genome encodes ten VAP27 homologs of which VAP27-1 and VAP27-3 are the most characterized isoforms to date[7,9,25,28,29]. VAP27 proteins have been localized to autophagosomes and to ER MCSs with the PM and mitochondria[4,11,25,29].

In non-plant species, VAPs interact with oxysterol-binding protein (OSBP) and its homologs, designated as OSBP-related (ORP) or OSBP-like (OSBPL) proteins[30]. These are small soluble proteins that bind sterols and other membrane lipids, including phosphatidylinositols (PIPs)[25,26]. A major function of ORPs is to transfer their lipid ligands between cellular membranes[26], including the counter-transport of PIPs and sterol, a process linked to the ability of these proteins to bind either lipid type in their well-conserved lipid-binding domain, the oxysterol-binding protein-related domain (ORD)[31]. The Arabidopsis genome encodes 12 ORPs[32,33]. Among these, ORP3A, a bona fide β-sitosterol-binding protein, localizes to the ER through an interaction with VAP27-3[28]. Furthermore, ORP2A was found to interact with PIP and VAP27-1 at the ER-PM MCSs and autophagosomes[4,29]. These results support an equivalence of ORPs across kingdoms for binding lipids and VAP27 proteins.

Here, we addressed the long-standing question on the composition of PLAMs and report the identification and the characterization of a novel protein complex at these sites. Based on their ubiquitous distribution at ER MCSs across kingdoms, we hypothesized that VAP27 proteins could also be constituents of the PLAMs. Consistent with our hypothesis, we demonstrated that VAP27-1 and VAP27-3 mark an ER subdomain where they interact with the phytosterol-binding protein ORP2A and physically associate with the chloroplast outer envelope membrane (OEM). Furthermore, we found that ORP2A and VAP27 proteins bind MGDG. Through lipidomic and cell biology analyses of loss-of-function mutants of single and high-order knock-out (KO)

alleles of *vap27-1*, *vap27-3*, and *orp2a*, we also demonstrated that the loss of the VAP27-ORP2A complex leads to a subtle alteration of the acyl composition of MGDG and PG as well as increased levels of phytosterols specifically and only in chloroplasts. Therefore, this work identifies components of the PLAMs and provides experimental evidence for a functional protein complex bridging the ER–chloroplast interface that is involved in aspects of lipid homeostasis maintenance in chloroplasts.

## Results

### The ER membrane proteins VAP27-1 and VAP27-3 mark an ER subdomain that is in close association with chloroplasts

Because VAP27 proteins are major constituents of the ER MCSs with PM and mitochondria[4,11,25,27,29], we investigated whether VAP27 proteins could also be part of the ER MCSs with chloroplasts. We first used live-cell confocal microscopy imaging for the analysis of functional fluorescent protein fusions to VAP27-1 and VAP27-3 (VAP27-1-YFP and VAP27-3-YFP)[4,9,11,27,29]. When closely inspecting the ER network of tobacco epidermal cells transiently expressing these constructs, along the cortical ER we observed a marked distribution of the VAP27 fluorescent protein fusions at ER sites with apparent higher levels of fluorescence relative to the bulk ER (Fig. 1, arrowheads). Such a distinctive distribution of fluorescence in the cortical ER was not as evident for the bulk ER membrane marker Calnexin (GFP-Calnexin)[34], which is an ER-targeted fusion of GFP to the transmembrane domain and cytosolic tail of the ER lumenal chaperone. GFP-Calnexin appeared distributed uniformly to the cortical ER network (Fig. 1, arrow). By monitoring chlorophyll natural fluorescence, which highlights chloroplasts, we found that VAP27 protein fusions accumulated in ER in close proximity to the chloroplasts. (Fig. 1, arrowheads). We named these areas ER membrane proximal to chloroplast (EMPC), for simplicity. Measurements of the fluorescence pixel intensity of the VAP27 protein fusions confirmed our qualitative observations of higher levels of fluorescence signal at the EMPC (Fig. 1; arrowheads) compared to the lower fluorescence signal at the bulk ER (Fig. 1, arrows), and of a relatively homogeneous distribution of the GFP-Calnexin signal throughout the cortical ER network (Fig. 1), supporting a distinctive accumulation of VAP27 proteins at the EMPC. By imaging multiple focal planes in the same cell and compiling the individual focal planes in Z-stacks as maximum projections, we verified that the distribution of the VAP27 at the EMPC occurred at the chloroplasts and appeared to be more significant to these proteins compared to other ER-associated proteins. Indeed, in the maximum projection imaging of several focal planes, the signal from GFP-Calnexin at the chloroplast was lower than VAP27, and the known ER-PM marker SYTA-YFP[35] exhibited a distinct distribution to punctate structures, which had a negligible overlap with the chlorophyll signal (Supplementary Fig. 1A). These qualitative observations were confirmed by measurements of the fluorescence signal overlap using Pearson's correlation coefficient analyses[36], which indicated significantly more positive values for the VAP27 proteins, and a hence higher signal overlap, compared to GFP-Calnexin and SYTA-YFP (Supplementary Fig. 1A).

Concomitant introduction of the ER lumen marker ER-mCherry[37] with the VAP27 protein fusions resulted in ER-mCherry detection at the EMPC (Supplementary Fig. 1B), indicating that the EMPC are continuous with the bulk ER. The distribution pattern of VAP27 proteins and GFP-Calnexin with respect to chloroplasts was reproducible in stable Arabidopsis transformants expressing these proteins under the control of a constitutive promoter (i.e., CaMV35S; Fig. 2a). VAP27 protein fusions expressed in stable Arabidopsis transgenics under the control of the respective endogenous promoters (Supplementary Fig. 1C) also marked EMPC, which showed a more punctate appearance compared to EMPC visualized by VAP27 proteins expressed transiently or under the control of a constitutive promoter (Figs. 1 and 2a and Supplementary Fig. 1), most likely because of different levels of

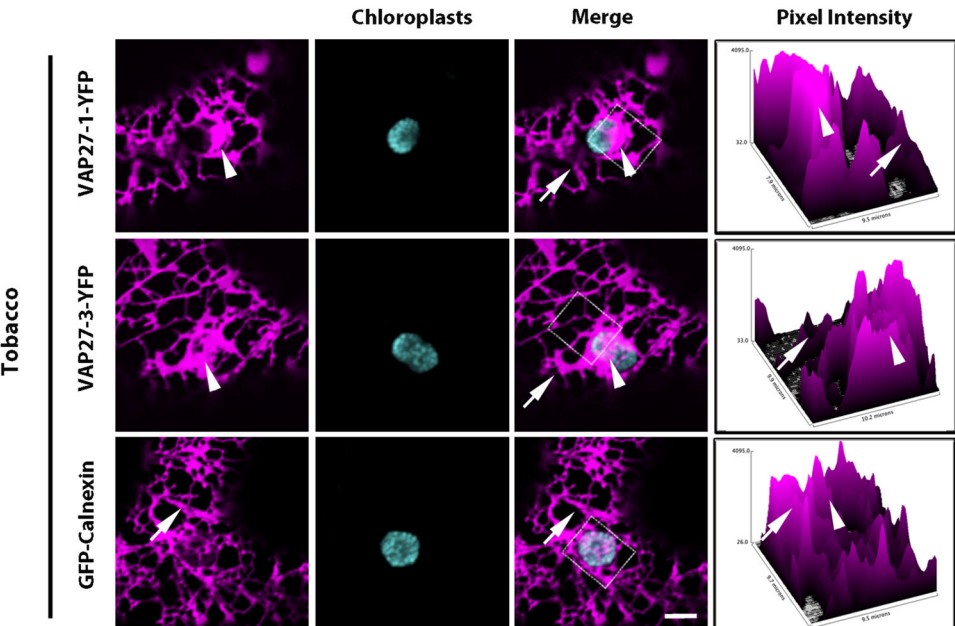

**Fig. 1 | VAP27-1 and VAP27-3 mark the bulk ER and accumulate in distinct ER regions.** Confocal images of live tobacco leaf epidermal cells transiently expressing VAP27-1 and VAP27-3 YFP fluorescent fusions showing a distribution to the tubules of the bulk ER (arrows) and bright fluorescent regions of the cortical ER (arrowheads), which are in close proximity to chloroplasts visualized through chlorophyll autofluorescence. A GFP-Calnexin fusion is distributed in the tubules of the bulk ER (arrows) with no marked accumulation in ER domains proximal to chloroplasts.

Measurements of the pixel intensity of the signal of the fluorescent protein fusions (right plots) show higher fluorescence intensity at the EMPC (marked by an arrowhead) compared to the bulk ER for the VAP27-fusions but not for GFP-Calnexin whose signal appeared more uniform in the ER network. In the pixel intensity plots, arrowheads and arrows point to the respective markers on the merged confocal images. Scale bar = 5 μm. Measurements from three independent experiments.

expression of the transgenes. These data support an inter-species conservation of the subcellular distribution of VAP27 proteins in the membrane of the ER, as also verified for ER-PM contact sites[27], and they also show conservation of the subcellular localization of these proteins at the ER–chloroplast interface.

To gather additional insights on the EMPC, we carried out fluorescence recovery after photobleaching (FRAP) assays on the pool of VAP27-YFP fusions at the EMPC and on the tubules of the bulk ER. We found that the half-time fluorescence recovery time of both VAP27-protein fusions was significantly higher at the EMPC compared to recovery in the ER tubules (Fig. 2b). These results support a reduced mobility of VAP27 proteins in the EMPC, which most likely underpins their preferential accumulation at these sites compared to the bulk ER. Furthermore, using the same approach, we compared the dynamics of GFP-Calnexin at the ER tubules and at the ER membrane proximal to chloroplasts. We found no differences in the half-time recovery and the mobile fraction for this marker (Fig. 2b), supporting the existence of mechanisms underpinning the increased residency of VAP27 proteins at the EMPC compared to our control (i.e., an ER membrane-associated protein marking the bulk ER).

We hypothesized that the accumulation of VAP27 proteins at the EMPC marked areas of close juxtaposition of the ER membrane with the chloroplast outer envelope membrane (OEM). To test this, we adopted a bimolecular fluorescence complementation (BiFC) approach[38,39]. In BiFC assays, YFP reconstitution can occur when the split YFP fragments are in close vicinity resulting in non-specific protein-protein interactions generated by the spontaneous reconstitution of the fluorochrome[39–42]. Therefore, we carried out BiFC assays with full-length VAP27 proteins fused to the C-terminal half of YFP (cYFP) (cYFP-VAP27). We co-expressed these constructs with Outer Envelope Protein 7 (herein OEP7; locus: *At3g52420*), a 7 KDa protein that is localized to the OEM through a membrane targeting determinant at the N-terminus[43]. We fused the C-terminus of OEP7 with the complementary N-terminal half of YFP (nYFP) (OEP7-nYFP). This fusion

results in the orientation of the C-terminal region of OEP7 in the cytosol, as supported by reconstitution of YFP fluorescence in cell expressing an untargeted (i.e., cytosolic) cYFP and OEP7-nYFP (Supplementary Fig. 2A). We expected that reconstitution of YFP fluorescence would occur if the ER membrane and the OEM were sufficiently close to favor spontaneous interactions of the complementary half YFP fusions to VAP27 and OEP7, respectively. To test this, we used transient expression in tobacco leaf as customary for this type of assays in plant cells[40]. We detected a reconstituted YFP fluorescence signal in sub-regions of the ER juxtaposed to chloroplasts (Fig. 3), supporting that the ER membrane marked by the VAP27 proteins is in close physical proximity with the OEM. As a negative control we used the ER integral membrane protein Root Hair Defective 3 (RHD3)[44,45] fused to cYFP at the N-terminus (cYFP-RHD3). The N-terminus of RHD3 is exposed in the cytosol[44], as also confirmed by reconstitution of fluorescence when cYFP-RHD3 was co-expressed with untargeted nYFP (Supplementary Fig. 2A). When cYFP-RHD3 was co-expressed with OEP7-nYFP, we did not verify fluorescence signal (Supplementary Fig. 2B), further supporting that the VAP27 proteins mark an ER domain proximal to chloroplasts that exhibits unique features compared to the bulk ER.

Together, the evidence provided thus far supports that VAP27 proteins mark an ER membrane subdomain interfacing with the chloroplasts, which is continuous with the bulk ER and exhibits unique properties favoring the preferential accumulation of VAP27-proteins and restricting their mobility at these sites.

## VAP27-1 and VAP27-3 interact with ORP2A at the EMPC

To gather functional insights into the EMPC, we conducted a genome-wide yeast two-hybrid (gwY2H) screen using the cytosolic domains of VAP27-1 and VAP27-3 (VAP27ΔTMD; i.e., VAP27 proteins devoid of the transmembrane domain, TMD, as required by the classic Y2H methodology) and an Arabidopsis seedling cDNA library. In the screen, we identified ORP2A as one of the most abundant hits (>20 hits). We confirmed the gwY2H interaction of ORP2A with VAP27-1 and VAP27-3

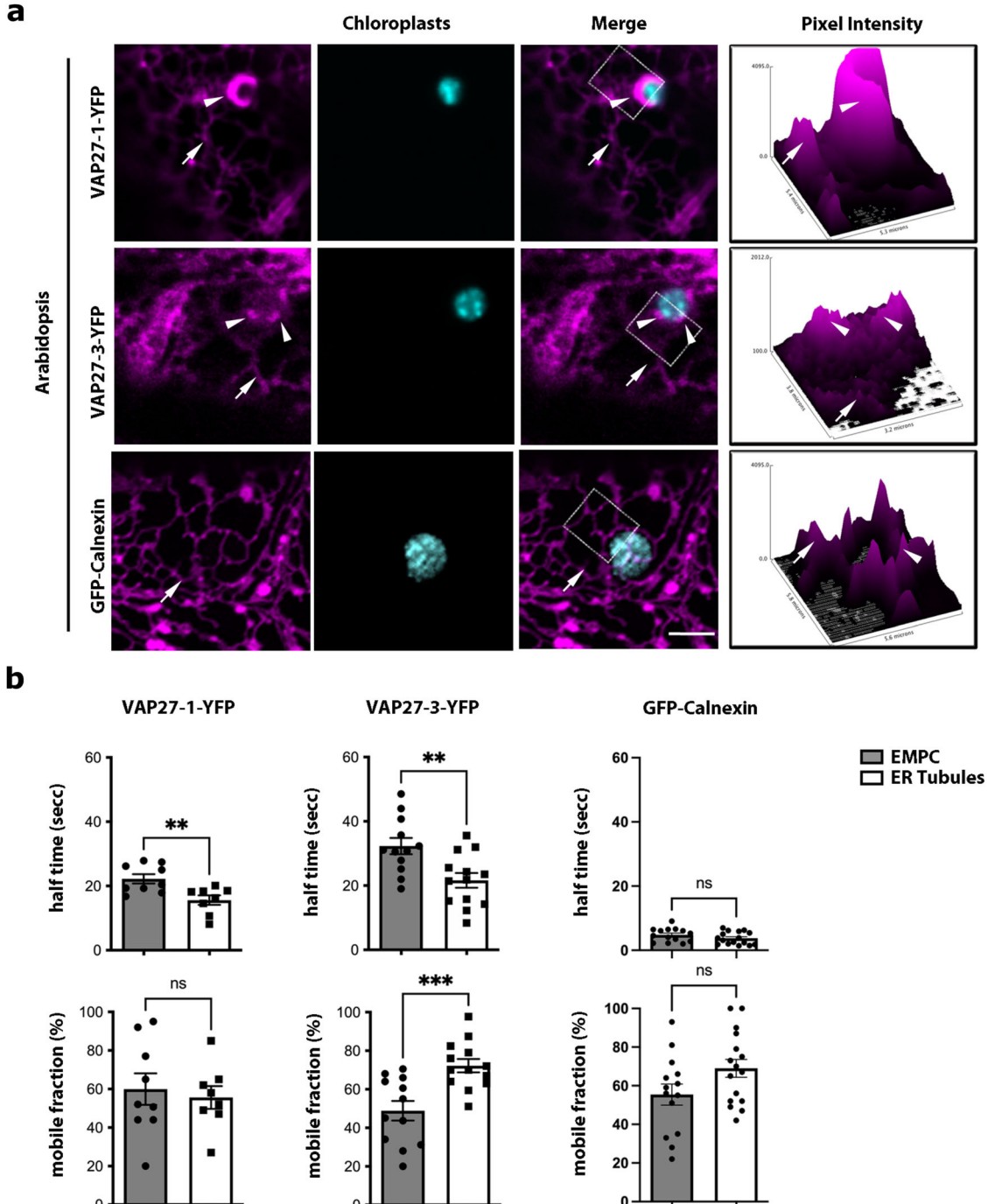

**Fig. 2 | VAP27-1 and VAP27-3 delineate an ER subdomain where they dynamically accumulate. a** Confocal images of live Arabidopsis leaf epidermal cells stably expressing VAP27-1 and VAP27-3 fluorescent fusions imaged to visualize YFP and chlorophyll autofluorescence showing that VAP27 proteins visibly accumulate at the EMPC (arrowheads). Such a distribution was not verified for GFP-Calnexin, which accumulates to the bulk ER (arrows). Measurements of pixel intensity of the signal of the VAP27 fluorescent protein fusions (right plots) show higher intensity at the EMPC (marked by an arrowhead) compared to the bulk ER (arrows) for the VAP27-fusions, while the distribution of GFP-Calnexin in the ER tubules (arrow)

appeared comparable to the ER at the region proximal to chloroplasts. In the pixel intensity plots, arrowheads and arrows point to the respective markers on the merged confocal images. Scale bar = 5 μm. **b** FRAP analyses on the pool of VAP27-1-YFP ($n = 9$) and VAP27-3-YFP ($n = 12$) at the EMPC show reduced mobility of VAP27 proteins at these sites (i.e., higher half-time recovery) compared to the protein pool at the ER tubules ($n = 8$ and $n = 13$, respectively) and compared to the control protein GFP-Calnexin ($n = 14$ for EMPC and $n = 16$ for ER tubules). Error bars indicate SEM. *$P < 0.05$, **$P < 0.01$, ***$P < 0.001$, ****$P < 0.0001$ ($P$ value calculated with one-way ANOVA with Tukey's post test).

through direct Y2H assays using VAP27ΔTMD as baits and full-length ORP2A as prey (Fig. 4a). The positive VAP27 protein-ORP2A interaction detected with Y2H was confirmed with in vitro pulldown analyses using recombinant VAP27ΔTMD fused to GST and full-length ORP2A fused to His (ORP2A-His; Fig. 4b). Together these results indicate that VAP27 proteins and ORP2A can interact with each other, as also recently

shown[29]. Based on these interaction results and the evidence that VAP27 proteins accumulate to the EMPC (Figs. 1–3 and Supplementary Fig. 1), we then aimed to investigate the subcellular distribution of ORP2A, and we generated stable Arabidopsis lines co-producing VAP27-CFP and ORP2A fused to YFP (ORP2A-YFP). Through imaging analyses of the cytoplasm, we found that the signal of ORP2A-YFP was

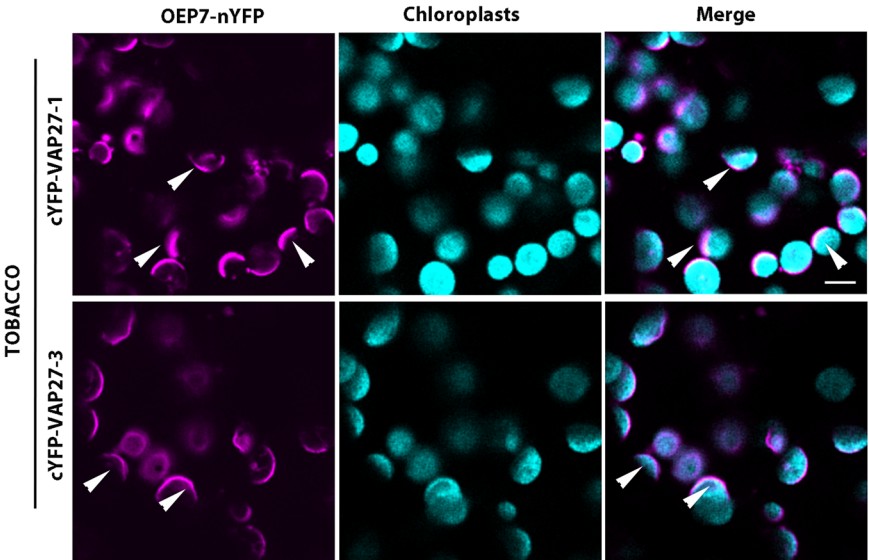

**Fig. 3 | The EMPC marked by VAP27 proteins are adjacent to chloroplasts.** Confocal images of live tobacco leaf epidermal cells expressing VAP27 and OEP fused to complementary halves of YFP (i.e., cYFP and nYFP). By using settings to visualize YFP and chlorophyll autofluorescence, BiFC analyses of cYFP fusions to VAP27-1 and VAP27-3 and OEP-nYFP show reconstitution of YFP fluorescence at the VAP27-EMPC only (arrowheads) indicating that the ER subdomain marked by VAP27 proteins is in sufficient proximity with the chloroplast OEM to facilitate YFP reconstitution. Scale bar = 5 μm. The images are representative of the observation coming from at least three independent experiments.

localized to the cytosol and chloroplasts, where it overlapped with the signal of VAP27 protein fusions at the EMPC (Fig. 4c).

Isolation and analyses of the *orp2a-1* knock-out allele (Supplementary Fig. 3A, B) showed that, compared to WT, the mutant exhibited developmental defects such as reduced size of the rosette (Supplementary Fig. 3C), indicating that the loss of *ORP2A* impairs growth. Furthermore, a triple mutant that we generated by combining *orp2a-1* with an established *vap27-1/vap27-3* double mutant[4] (*vap27-1/vap27-3/orp2a-1*) displayed a similar growth phenotype to *orp2a-1* (Supplementary Fig. 3C). Also, a quantification of the size of the major axis of mesophyll cells of the mutants showed a significantly lower length compared to the WT but a similar size of *orp2a-1* compared to *vap27-1/vap27-3/orp2a-1* (Supplementary Fig. 3D). Together these results support that ORP2A, VAP27-1, and VAP27-3 are genetically linked in processes regulating plant and cell development.

### ORP2A and VAP27 proteins associate with the OEM where they interact with MGDG

We next aimed to gain insights into the distribution of ORP2A and VAP27 proteins with respect to the OEM. To do so, we carried out protein association assays using isolated highly purified intact pea chloroplasts and in vitro transcribed and translated radiolabeled ORP2A-His ($^3$H-ORP2A), followed by trypsin digestion (Fig. 4d). As controls, we used luciferase to assay for non-specific binding of a protein to the OEM, and the Translocon Outer Envelope 33 (Toc33), a single membrane-spanning protein that is integrated in the chloroplast OEM with the bulk of the protein facing the cytosol[46]. With its subcellular distribution and protein topology, Toc33 served to validate our confocal microscopy observations on the distribution of ORP2A to the chloroplasts and gain topological insights on the association of ORP2A with the OEM. Fluorograms were used to track the association of either $^3$H-ORP2A, $^3$H-Luciferase, or $^3$H-Toc33 to the chloroplast OEM. Upon trypsin digestion, we found that trypsin completely digested both Toc33 and ORP2A, supporting that ORP2A is localized to the OEM, whereas the stroma localized small subunit of Rubisco (SSU) was completely protected from tryptic digestion (Fig. 4d, upper panels), thus confirming the integrity of the chloroplasts used in our association assay. The radiolabeled luciferase control, which does not have a transit peptide, did not associate with chloroplasts, and was entirely digested by trypsin (Fig. 4d, upper panels), confirming specificity of our association assay. These results are in accordance with our live-cell imaging results for a localization of ORP2A at the chloroplasts (Fig. 4c) but also indicate that ORP2A associates to the OEM and that the topology of ORP2A association with the chloroplasts is such that it binds to the OEM facing the cytosolic side.

The experiments were repeated with $^3$H-ORP3A, a member of the ORP family that has been shown to interact with VAP27-3 at the bulk ER[28]. We found that, unlike ORP2A, ORP3A did not associate with the OEM (Supplementary Fig. 6), indicating that the interaction of ORP2A with the OEM is specific to ORP2A.

We next extended our in vitro association assays to include in vitro transcribed and translated radiolabeled VAP27ΔTMD, $^3$H-VAP27ΔTMD. These truncated protein forms that lack the VAP27 protein transmembrane domain were used to increase recombinant protein solubility for purification and in vitro analyses. In the association reaction, $^3$H-VAP27ΔTMD was recovered with isolated chloroplasts but digested by trypsin (Fig. 5a, lower panels), supporting an association of VAP27 proteins to the OEM facing the cytosolic side similar to ORP2A (Fig. 5a, upper panels and Supplementary Fig. 6A, B).

Based on the evidence that VAP27-1 and VAP27-3 interact with ORP2A (Fig. 4a, b) and that VAP27 proteins and ORP2A contact the OEM (Figs. 4d and 5a), we next aimed to test whether VAP27 proteins and ORP2A could directly bind plastid lipids at the OEM. VAP27 proteins do not interact with PC nor PI[4], which can be found in the OEM[47]. Therefore, we spotted the galactolipids MGDG, DGDG, and PG, on membranes and incubated them with recombinant His-protein fusions to VAP27ΔTMD and ORP2A. As positive control, we used PI3P, which interacts with VAP27 proteins and ORP2A[4,29]. We established that VAP27-1ΔTMD, VAP27-3 ΔTMD, and ORP2A bind to MGDG but not PG and DGDG (Fig. 5b).

Taken together, the verified interaction of ORP2A and VAP27 proteins and the lipid interaction results support the likelihood that VAP27 proteins and ORP2A form a protein complex that associates with the OEM. These results also indicate that VAP27 protein and ORP2A can interact with OEM-specific glycerolipids directly. Because

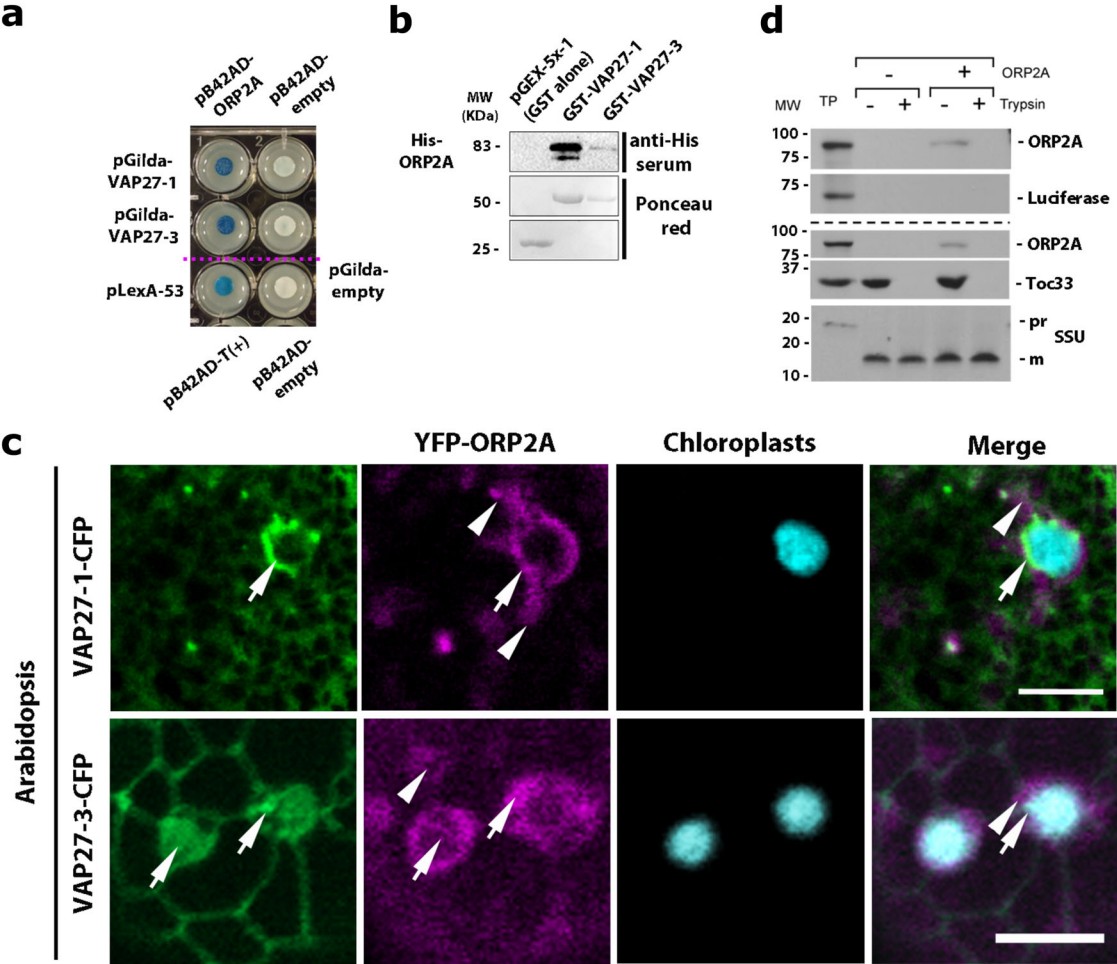

**Fig. 4 | VAP27 proteins interact directly and colocalize with ORP2A at the EMPC. a** Direct Y2H interaction assays between the cytosolic domain of the two VAP27 proteins and full-length ORP2A show interaction between these proteins. pLexA-53 and pB42AD-T(+) positive control; pB42AD-empty and pGilda-empty, negative controls. **b** Western blot analyses with anti-His serum on eluates GST-column of columns loaded with GST-VAP27 and His-ORP2A fusions show a direct interaction of VAP27 proteins and ORP2A. Ponceau red-stained gels show the respective protein loading. Three independent experiments were conducted. **c** Confocal live-cell imaging of Arabidopsis stable transformants co-expressing CFP fusions to VAP27 proteins and YFP-ORP2A show that ORP2A and VAP27 proteins colocalize at the EMPC (arrows). ORP2A also localizes to the cytosol (arrowheads). Scale bars = 5 μm. **d** Upper panel. SDS-PAGE and fluorography analysis of in vitro association assays using pea chloroplasts show that [³H]ORP2A was recovered only

in the absence of trypsin treatment, while luciferase, which lacks a chloroplast targeting signal, did not associate with chloroplasts, thus supporting a specific association of ORP2A with chloroplasts. Lower panel. SDS-PAGE and fluorography analysis of the control proteins, Toc33 (OEM (outer envelope membrane) protein) and SSU (small subunit of rubisco protein located within the stroma) demonstrated that the chloroplasts used in this assay were intact based on the evidence that similar to [³H]ORP2A, [³H]Toc33 associates with the OEM but is subsequently digested by trypsin whereas the stroma control protein, [³H]SSU is imported into the chloroplast stroma and is not digested by trypsin. Therefore, we concluded that ORP2A is associated with the OEM. TP 10% of translation product added to association assay, pr precursor, m mature forms of the Small Subunit of Rubisco. Three independent experiments were conducted.

the single VAP27 transmembrane domain is necessary for the anchoring of these proteins to the ER membrane[25,27,28], the association of VAP27 proteins with the OEM is most likely direct and peripheral. Next, to test the effect of the loss of ORP2A, VAP27-1, and VAP27-3 on ER–chloroplast association, we measured the fluorescence overlap between the ER labeled with the lipophilic dye for the ER membrane, DiOC$_6$[48], with the chlorophyll autofluorescence in WT and the mutants *orp2a-1*, *vap27-1/vap27-3*, and *vap27-1/vap27-3/orp2a-1* (Supplementary Fig. 4). A Pearson's correlation coefficient analysis of the degree of overlap between the ER membranes and the chlorophyll auto-fluorescence showed that the loss of VAP27 proteins and ORP2A in the respective double, single and triple mutant combinations resulted in a small but similarly reduced correlation coefficient compared to WT (Supplementary Fig. S4). These results indicate that the cellular availability of VAP27 proteins and ORP2A is necessary for the homeostasis of the ER–chloroplast interactions.

## The loss of VAP27 and ORP2A affects plastid lipid homeostasis in subtle ways

Because of the verified interaction of VAP27-1, VAP27-3, and ORP2A with MGDG (Fig. 5b), we next aimed to investigate a possible involvement of VAP27-1, VAP27-3, and ORP2A in chloroplast lipid trafficking or homeostasis using WT and loss of function mutants of VAP27 and ORP2A. Therefore, we tested the lipid composition of the leaf tissue of WT, *orp2a-1*, *vap27-1/vap27-3* double mutant[4], and the *vap27-1/vap27-3/orp2a-1* triple mutant. The lipid composition was analyzed as described earlier[49]. Unlike bona fide lipid trafficking mutants, we only observed subtle differences in the total acyl composition across the mutants compared to WT (Fig. 6). Specifically, the acyl compositions of PE and PI, which are primarily ER lipids, and of PC, which is also found in the OEM, were not drastically altered in the mutants. However, we found an increase in the relative levels of 18:1 (carbons: double bonds) and 16:0 in PG of *vap27-1/vap27-3/orp2a-1*, and a decrease of

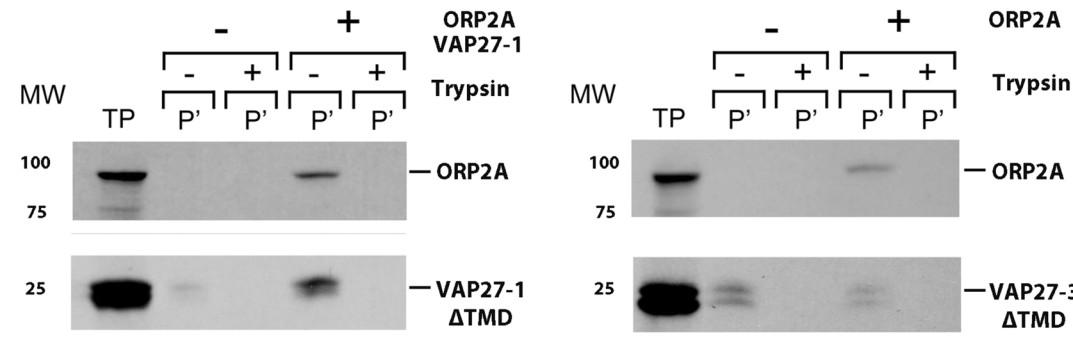

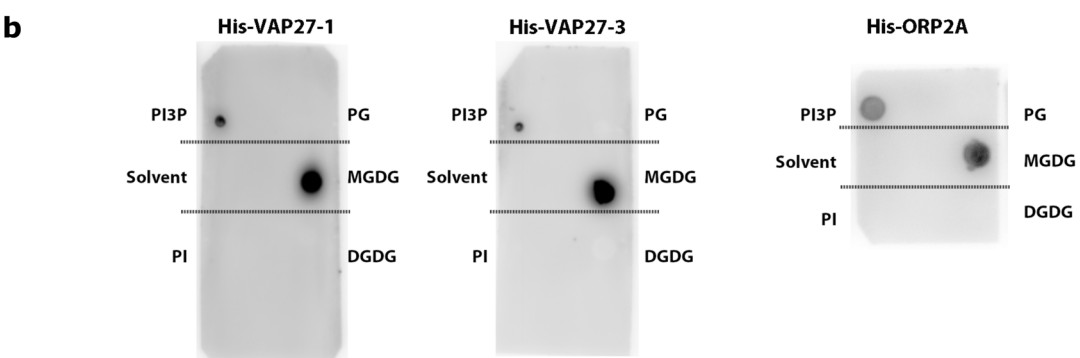

**Fig. 5 | ORP2A and VAP27 proteins associate with the OEM and OEM lipids. a** In vitro association assays with isolated pea chloroplasts incubated with either [³H] ORP2A (upper panel) and [³H]VAP27-1ΔTMD or [³H]VAP27-3 ΔTMD (lower panels). The resulting fluorograms show that both [³H]ORP2A (upper panel) and [³H]VAP27-1 ΔTMD or [³H]VAP27-3 ΔTMD (lower panels) associate with OEM but are susceptible to trypsin digest. Hence, VAP27 proteins associate with OEM, similarly to ORP2A. Three independent experiments were conducted. **b** Protein–lipid overlay assay with PG, MGDG, and DGDG with recombinant His-tag fusions to ORP2A, VAP27-1ΔTM, and VAP27-3ΔTM revealed with immunoblot with anti-His serum. The assays indicate that ORP2A, VAP27-1ΔTM, and VAP27-3ΔTM interact directly with MGDG. Lipid solvent and PI were used as negative controls; PI3P was used as positive control.

18:3 in PG of the *vap27-1/vap27-3* and *vap27-1/vap27-3/orp2a-1* mutants. Furthermore, reduced levels of 18:2 in SQDG were found in the *vap27-1/vap27-3* and *vap27-1/vap27-3/orp2a-1* mutants. Interestingly however, markedly increased levels of 16:1 and 18:1 in MGDG were observed in all three mutant backgrounds compared to WT. While these differences were not indicative of changes in the relative contributions of the ER or the plastid assembly pathways per se, they indicate that the loss of VAP27 proteins and ORP2A affects the acyl composition of thylakoid lipids that are synthesized uniquely, i.e., MGDG and SQDG, or primarily, i.e., PG, in chloroplasts.

Even very small changes in the homeostasis of polar lipids in thylakoids have been associated with alteration at thylakoidal membrane organization in chloroplasts[50,51]. Therefore, we next aimed to analyze the organization of thylakoids using transmission electron microscopy of *orp2a-1*, *vap27-1/vap27-3*, and *vap27-1/vap27-3/orp1a-1* mutants for comparison to WT. We found that the organization of the thylakoids was different in *orp2a-1*, *vap27-1/vap27-3*, and *vap27-1/vap27-3/orp1a-1* mutants compared to WT. Specifically, when we measured number of grana per chloroplast, as well as grana length and thickness, and lamellae thickness, we established that, compared to WT, the mutants exhibited similar lamellae thickness but also an overall reduced grana size and an higher number of grana per chloroplast (Supplementary Fig. 5). These results combined with the observed plant and cell phenotypes (Supplementary Fig. 3C, D) indicate that the mutations of VAP27 proteins and ORP2A lead to significant alterations of the

photosynthetic membranes of chloroplasts, which most likely affect plant development.

### ORP2A and VAP27 proteins are necessary for sterol homeostasis in chloroplasts

Based on the reported interaction of ORPs with sterols in several species, including plants[28,52–54], we next aimed to test a possible involvement of ORP2A in sterol homeostasis. Sterol biosynthesis occurs in the ER[55], and in plants the predominant forms of sterols are β-sitosterol, stigmasterol, and campesterol[56,57]. Within the endomembrane system, the PM has the highest sterol-to-phospholipid molar ratio[56,58]. Sterols have been detected also in the tonoplast, mitochondria, and chloroplasts[58–60]. Arabidopsis mutants for key regulators of the sterol biosynthetic pathway exhibit severe phenotypes, including lethality[61], and altered chloroplast development[62], supporting a critical role for sterols in plant physiology. For our analyses, we first took a pharmacological approach and tested the response of WT, *orp2a-1*, *vap27-1/vap27-3*, and *vap27-1/vap27-3/orp2a-1* to lovastatin 300 nM (LOVA300), an inhibitor of the sterol biosynthetic pathway[63], known to cause a reduction of root growth in Arabidopsis[64]. When grown on control media containing DMSO (lovastatin solvent), the mutant lines displayed a reduced root growth phenotype compared to the WT (Fig. 7a). Upon lovastatin treatment, both WT and the mutant lines exhibited a reduction of root elongation, but this effect was particularly noticeable for the mutants (Fig. 7a). Estimation of the ratio of the root growth measurements (DMSO/lovastatin ratio) confirmed that

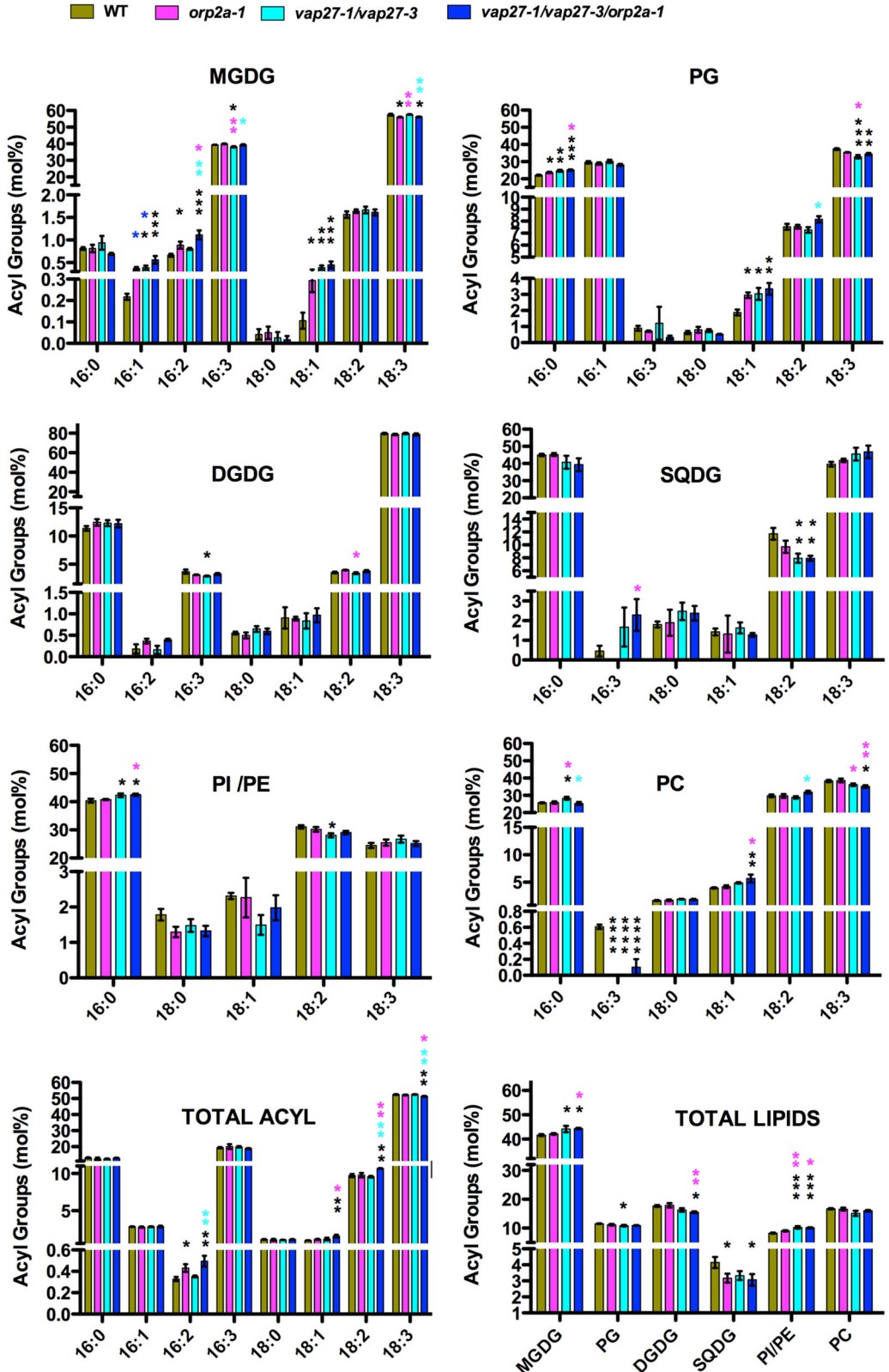

**Fig. 6 | Profiling of cellular lipid shows small but significant impact of a lack of ORP2A and VAP27 proteins.** Graphical representation of acyl groups quantification (mol%) for MGDG, PG, DGDG, SQDG, PI-PE, PC, total acyl, and total lipids, respectively in WT, *orp2a-1*, *vap27-1/vap27-3*, and *orp2a-1/vap27-1/vap27-3* (for all the lines, *n* = 4). Error bars indicate SEM. *P < 0.05, **P < 0.01, ***P < 0.001, ****P < 0.0001 (P value calculated with one-way ANOVA with Tukey's multiple comparison test).

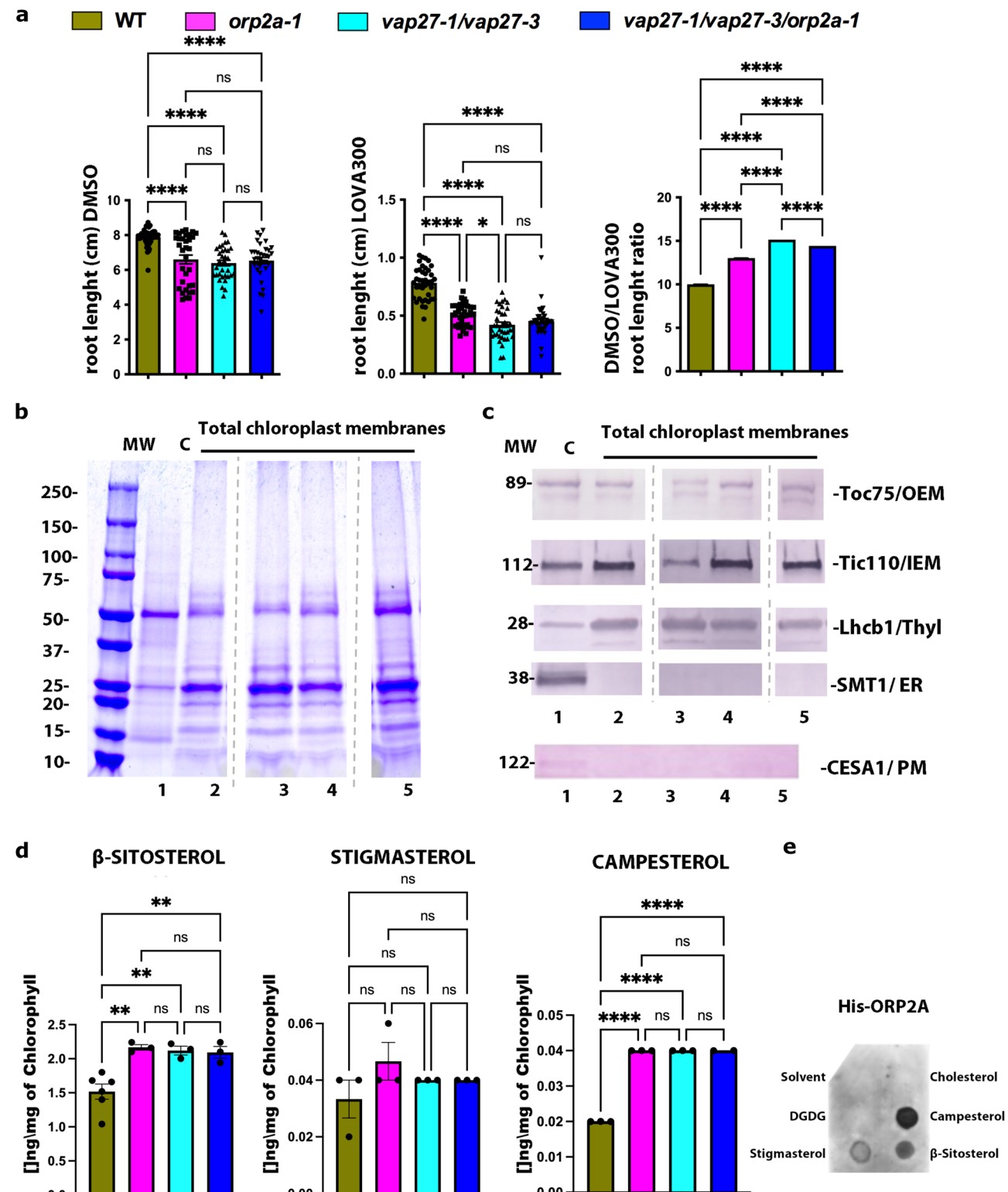

the mutants exhibited higher growth inhibition levels in the presence of lovastatin compared to WT (Fig. 7a). These results suggested a potential involvement of VAP27 proteins and ORP2A in sterol homeostasis and prompted us to test the role of VAP27 proteins and ORP2A in plastid sterol homeostasis. To do so, we isolated intact chloroplasts from WT, *orp2a-1*, *vap27-1/vap27-3*, and *vap27-1/vap27-3/orp2a-1* seedlings to compare the sterol levels in these organelles (Fig. 7b, c). We reasoned that working with purified chloroplasts would have allowed us to detect changes, if any, in sterols in the various

backgrounds without the interference of sterol-rich membranes (e.g., PM). We first ensured that we recovered all three chloroplast membrane compartments (i.e., OEM, inner envelope membrane, and thylakoids) by assaying for signal on Western blots with antibodies for Toc75 (OEM), Translocon Inner Complex 110 (Tic110) (IEM), and light-harvesting chlorophyll a/b binding protein (LHCPII) (thylakoids) (Fig. 7c). We also assayed for contamination of the isolated chloroplasts by non-PLAM ER membrane by probing with an antibody for the ER membrane-associated sterol methyltransferase SMT1[61]. Based

**Fig. 7 | VAP27 proteins and ORP2A are necessary for sterol homeostasis. a** Root growth measurements on 2-week-old seedlings grown on vertical plates containing either DMSO (negative control/ lovastatin solvent) or lovastatin (300 nM) show that single *orp2a-1*, double *vap27-1/vap27-3* and triple *vap27-1/vap27-3/orp2a-1* exhibit a reduced growth in the presence of the drug as confirmed by the calculation of the DMSO/lovastatin root length ratio. ($n = 43$ for WT, $n = 33$ for *orp2a-1*, $n = 34$ for *vap27-1/vap27-3*, $n = 34$ for *vap27-1/vap27-3/orp2a-1* in DMSO treatment), ($n = 39$ for WT, $n = 38$ for *orp2a-1*, $n = 37$ for *vap27-1/vap27-3*, $n = 39$ for *vap27-1/vap27-3/orp2a-1* in lovastatin treatment). Error bars indicate S.E.M (*P* value calculated with one-way ANOVA with Tukey's post test). **b, c** Proteins extracted from either total WT leaf (**c**) (lane 1) or total chloroplast membranes (5 μg chlorophyll/lane) isolated from the following backgrounds: WT (lane 2), *vap27-1/vap27-3* (lane 3), *orp2a-1* (lane 4), *vap27-1/vap27-3/orp2a-1* (lane 5) were either run on a Coomassie SDS-PAGE gel (**b**) or for Western blot analyses (**c**). For the Western blot, membranes were probed with antibodies for the OEM (Toc75), IEM (Tic110), thylakoids (LHCPII), and the ER

membrane protein SMT1 and PM marker CESA1. The analyses did not show the presence of ER membrane or PM contamination in the isolated chloroplast fraction. Total Western blot membrane is shown in Supplementary Fig. 7. **d** Measurements of β-sitosterol (wt: $n = 6$, *orp2a-1*: $n = 3$, *vap27-1/vap27-3*: $n = 3$, *vap27-1/vap27-3/orp2a-1*: $n = 3$, respectively), campesterol (wt: $n = 3$, *orp2a-1*: $n = 3$, *vap27-1/vap27-3*: $n = 3$, *vap27-1/vap27-3/orp2a-1*: $n = 3$, respectively) and stigmasterol wt: $n = 3$, *orp2a-1*: $n = 3$, *vap27-1/vap27-3*: $n = 3$, *vap27-1/vap27-3/orp2a-1*: $n = 2$, respectively, on equal amount of extracted chloroplast fractions show that single *orp2a-1*, double *vap27-1/vap27-3* and triple *orp2a-1/vap27-1/vap27-3* have altered ratio among the tested sterols per mg of chlorophyll/ FW. Error bars indicate SEM. **e** Protein–lipid overlay blot with recombinant ORP2A-His shows that the protein interacts with β-sitosterol, stigmasterol and campesterol and does not interact with cholesterol. DGDG was used as an additional negative control along with the sterol solvent control. *$P < 0.05$, **$P < 0.01$, ***$P < 0.001$, ****$P < 0.0001$ (*P* value calculated with one-way ANOVA with Tukey's post test).

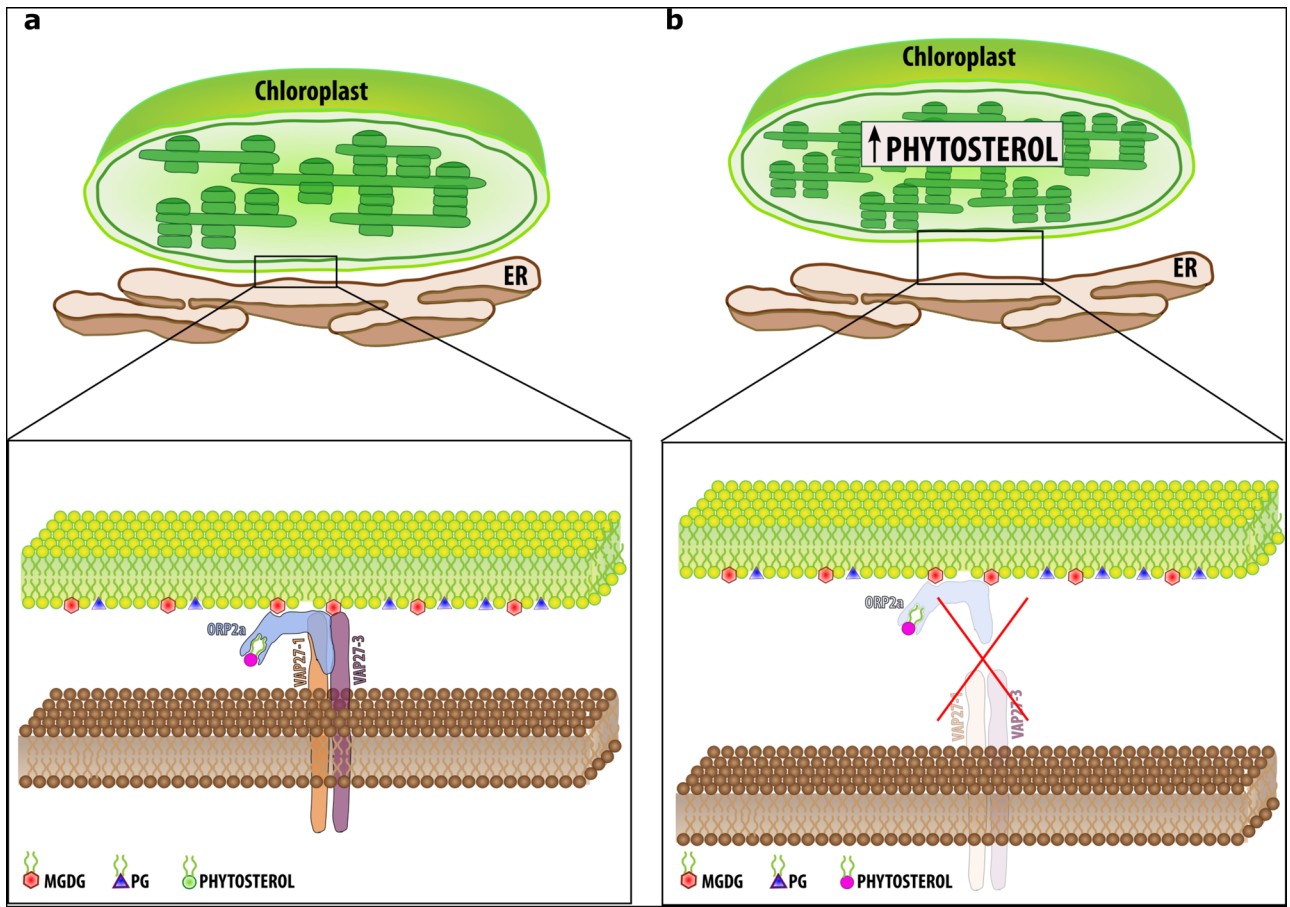

**Fig. 8 | Model for the organization and function of the VAP27-ORP2A PLAMs (voPLAMs).** Proposed model for the voPLAMs envisioning the localized presence of the VAP27 proteins at specific subdomains of the ER where they form a functional complex with ORP2A. VAP27 proteins preferentially accumulate at these sites, possibly through an interaction with the OEM. By means of their single transmembrane domain and a direct interaction with lipids (i.e., MGDG), which are present in the OEM, the VAP27 proteins bridge the ER with the OEM serving as an anchor between the membrane of the ER and chloroplasts. VAP27 and ORP2A interact with each other, and ORP2A associates directly with the OEM most likely via an interaction with MGDG. ORP2A also interacts with phytosterols. By interacting

directly with the OEM and VAP27 proteins, ORP2A is recruited to the voPLAMs where it likely serves to maintain homeostasis of phytosterols at the ER–chloroplast interface (**a**). The model summarizes the results obtained from the tested hypothesis: in the absence of the VAP27-1, -3 and ORP2A, the functional association of the ER and the OEM is at least partially disrupted, which in turn negatively affects homeostasis of phytosterols. Our results also indicate that the inner membrane morphology also changes in absence of VAP27-1, -3, and ORP2A (**b**), as a likely consequence of an altered phytosterol homeostasis at the ER/OEM interface. OEM chloroplast outer envelope membrane, ER endoplasmic reticulum, PG phosphatidylglycerol, MGDG monogalactosyldiacylglycerol.

on the absence of SMT1 signal in the Western blot analyses (Fig. 7b, c and Supplementary Fig. 7B–D), we concluded that ER membrane contamination of our purified chloroplast samples, if any, was very low. Because the PM is a site of accumulation of sterols, we also assayed the isolated chloroplasts for PM using an antibody for the

multi-spanning membrane protein cellulose synthase 1 (CESA1), which is embedded in the PM[65,66]. Similarly to the results for bulk ER contamination, we did not find evidence for PM contamination in the isolated chloroplast fraction (Fig. 7c and Supplementary Fig. 7B, D). We then measured the levels of β-sitosterol, campesterol and stigmasterol,

the most abundant sterols in Arabidopsis[67]. We found significantly higher levels of β-sitosterol and campesterol in the chloroplasts isolated from *orp2a-1*, *vap27-1/vap27-3*, and *vap27-1/vap27-3/orp2a-1* compared to WT (Fig. 7d). Interestingly the levels of phytosterol increases in the triple *vap27-1/vap27-3/orp2a-1* mutant were not statistically different from the single *orp2a-1* and double *vap27-1/vap27-3* mutant (Fig. 7d), supporting a possible functional interaction of these proteins in sterol homeostasis. An analysis of the total leaf sterol content in WT and the mutants did not show a significant difference across the backgrounds except for a small reduction in the levels of β-sitosterol in the triple *vap27-1/vap27-3/orp2a-1* mutant (6%; Supplementary Fig. 8). Because the levels of β-sterol and campesterol in chloroplasts are higher in the mutants compared to WT (Fig. 7d) but no differences in total sterols were verified across WT, *orp2a-1*, *vap27-1/vap27-3*, and only a slight reduction in β-sitosterol was verified in the *vap27-1/vap27-3/orp2a-1* triple mutant (Supplementary Fig. 8), the total sterol availability is unlikely causative of the observed increase in sterol levels in chloroplasts in the mutants compared to WT.

Based on the role of non-plant homologs of ORP2A in the intracellular transport of sterols[68] and the reported interaction of Arabidopsis ORP3 with VAP27-3[28], we next tested the interaction of ORP2A with β-sitosterol, campesterol and stigmasterol using spotted sterols on membranes and recombinant His-ORP2A. We found that ORP2A interacted with these phytosterols (Fig. 7e). Together, these results support that the VAP27 proteins and ORP2A are necessary to maintain sterol homeostasis in chloroplast membranes.

## Discussion

ER and chloroplasts are essential organelles that physically interact through largely unknown mechanisms. In this work, we provide evidence that two ER membrane-associated VAP27 proteins, VAP27-1 and VAP27-3, interact with ORP2A at ER subregions that interface with chloroplasts, where they preferentially accumulate and physically associate with the OEM. We also demonstrate that the loss of the VAP27 proteins and ORP2A alters the homeostasis of chloroplast sterol pools. Furthermore, our results show that VAP27 proteins and ORP2A bind lipid constituents of the OEM, and that ORP2A binds directly phytosterols. Taken together, our results indicate that VAP27 proteins and ORP2A mark an ER subdomain interfacing with the chloroplasts where they form a protein complex that interacts directly with OEM lipids and is necessary for the lipid/sterol homeostasis of the chloroplast membranes (Fig. 8). We propose that the ER subdomains identified and characterized in this work represent a PLAM subpopulation that we define as VAP27-ORP2A-PLAMs (voPLAMs), which are likely not directly involved in bulk lipid exchange as part of the ER pathway of chloroplast lipid assembly but are involved in maintenance of sterol homeostasis in chloroplasts.

In all eukaryotes, the ER pervades the cell and associates with most cellular compartments, such as mitochondria, peroxisomes, endosomes, and the PM through MCSs. These structures generally represent regions of close membrane apposition (i.e., within 10–20 nm) and play key roles in lipid transport and synthesis, calcium homeostasis, organelle positioning and dynamics[69]. In non-plant species, several ER tethers that underlie the MCSs through protein–protein or protein–membrane interactions are known[70,71], but knowledge about the MCS proteome is much more limited in plants[12]. Through early electron microscopy studies, the close physical apposition of the ER with the cytoskeleton and heterotypic membranes has been known for decades[14]. Similarly, the functional interaction of the ER with non-secretory organelles, such as chloroplasts for lipid production, has been known for a long time[72]. Nonetheless, prior to this work, the identity of the constituents of the ER–chloroplast MCSs was unknown. In this work, we show that VAP27-1 and VAP27-3 interact with the OEM and directly with MGDG, which are galactolipids unique to the plastid OEM. We also show that in addition to their distribution to the

bulk ER, VAP27 proteins accumulate at the voPLAMs where they exhibit reduced mobility. The mechanisms for the reduced mobility of VAP27 proteins at the ER-PM MCSs compared to an ER bulk membrane protein[27], and at the voPLAMs compared to the VAP27 pool in the bulk ER (this work) are yet unknown. The existence of a protein complex with VAP27 was hypothesized to be causative of the reduced mobility of VAP27 proteins at the ER-PM MCSs[27]. Similarly, it is possible that an interaction of VAP27 proteins with OEM proteins and/or simply a direct interaction with OEM lipids contribute to the reduced mobility in the ER membrane at the voPLAMs and a preferential yet dynamic accumulation of these proteins at these sites. Based on these findings, we propose that VAP27 bridge the ER membrane subdomains, where they accumulate, by binding directly with the OEM (Fig. 8).

The ten plant VAP27 proteins, three yeast Scs2, and two animal VAP proteins constitute a highly conserved family of integral membrane proteins in eukaryotes[73]. Through high-throughput screens, the animal VAPA and VAPB have been shown to possibly interact with more than 250 proteins, 50% of which are common to both proteins and are involved, or predicted to be involved, in lipid transfer between organelles[74]. Similarly, plant VAP27-1 and VAP27-3 have a large spectrum of interactors[4,11,12,27,28,75,76], supporting conservation of the interaction of VAP27 homologs with multiple proteins involved in diverse biological pathways. In this work, we support recent findings that VAP27-1 interacts with ORP2A[29], and we demonstrate that the interaction is extended to VAP27-3, adding to the list of known interactors of the two proteins. The evidence provided in this work that VAP27 proteins interact with ORP2A at the voPLAMs supports that these proteins may have assumed features to suit metabolic and physiological demands of photosynthetic cells. This is further supported by our findings that a loss of ORP2A leads to an altered organization of chloroplasts in mesophyll cells compared to WT.

While the plasticity of the VAP27-protein interaction with lipids is likely pivotal to their function as a platform for an interaction of the ER with heterotypic membranes that have a diverse composition, the interaction of VAP27 proteins with their protein partners confers them the ability to take part in many cellular processes, such as membrane traffic, lipid transport, calcium homeostasis and the unfolded protein response[74]. For instance, in agreement with a demonstrated direct interaction of VAP27 proteins with lipids enriched in endocytic membranes and clathrin, the loss of VAP27-1 and VAP27-3 compromises endocytosis[4]. In our work, we demonstrate that VAP27-1 and VAP27-3 directly bind the OEM-specific lipid MGDG and the phytosterol-binding protein ORP2A, and that their loss leads to altered sterol content in the chloroplasts, in agreement with a role of VAP proteins to function as a platform for cellular processes that involve their interacting proteins. On the other hand, ORP2A has been previously shown to bind phospholipids[29], and the evidence provided in our work that ORP2A binds phytosterols (i.e, MGDG), supports the ability of this protein to interact with several lipid types, including lipids of photosynthetic organelles, underscoring that the cellular function of the VAP27-ORP2A complex is likely broad. MGDG and DGDG constitute ~50 and ~30% of chloroplast membrane lipids, respectively[77]. Structurally, the VAP proteins, including the yeast Scs2p/22p, have three major domains: (i) a MSP domain that is capable of binding proteins containing the two phenylalanine residues in an acidic tract (FFAT) motif; (ii) a C-terminal domain that spans the ER membrane; (iii) an intermediate variable linker region[78]. Scs2p/22p have both been shown to bind PIPs through their MSP domain as a stringent requisite for their function[79]. Similarly, plant VAP27 proteins have been shown to interact with a variety of lipids, including PIPs[4,29]. Our work adds to these findings by showing a direct interaction with MGDG. Although the lipid-binding pocket has not been mapped on VAP27 proteins, the ability of these proteins to bind lipids has been demonstrated with recombinant proteins devoid of the transmembrane domain. Therefore, the VAP27 lipid binding occurs on either the MSP domain or the

intermediate variable linker region. Based on the conserved structure of the VAP proteins the lipid binding of VAP27 proteins, including the binding to MGDG, may occur at the MPS similar to the yeast counterpart, but this is still to be experimentally tested.

The most characterized function of non-plant VAPs to date concerns the transport of lipids from the ER to their destination organelle via interacting lipid transfer proteins. For example, CERT (STARD11) interacts with VAPs, extracts ceramide from the ER membrane and transfers it to phosphatidylinositol-4-phosphate of the *trans*-Golgi membrane[80]. Similarly, Nir2 interacts with VAPA/VAPB and transfers phosphatidylinositol from the ER to the PM[81]. In our work, we have seen small but significant differences in the cellular levels of PG and MGDG between WT and deletion mutants of VAP27 proteins and ORP2A. Although these results imply that VAP27 proteins and ORP2A are necessary to maintain the homeostatic levels of these lipids, they do not allow us to conclude a role of VAP27 proteins and ORP2A in chloroplast lipid import as part of the ER pathway of lipid assembly. Similar to the yeast VAP homolog, which binds to lipid-binding proteins such as Osh2, Osh3 to the ER-PM interface, where they regulate lipid metabolism[78], the VAP27-ORP2A complex may have a regulatory role on the biosynthesis of MGDG and PG. Alternatively, the interaction of VAP27 proteins with MGDG may serve to stabilize an ER–chloroplast interaction, and a loss of the VAP27-ORP2A complex may compromise the functional interactions of ER and chloroplasts in lipid synthesis, resulting in an alteration of membrane lipid abundance.

Sterols are essential in eukaryotic membranes[82]. Plant sterol types are numerous with predominance of β-sitosterol, campesterol, and stigmasterol[83–85]. The biosynthesis of phytosterols has been defined at a biochemical level, and evidence based on biochemical fractionation studies and subcellular localization analyses of sterol biosynthesis enzymes support that the biosynthesis of sterols occurs at the ER[55,56,58,61,86]. Nonetheless, at steady state, sterols accumulate in the PM, supporting that the traffic of sterols from the ER is rapid[56,58,87]. How sterols are trafficked out of the ER is still largely unknown. While it is possible that traffic may follow the bulk flow of membrane to the PM via the conventional secretory pathway, it has been shown that VAP27-3 interacts with the sitosterol-binding protein ORP3A at the ER[28]. Abrogation of the VAP27-3-ORP3A interaction leads to a relocation of ORP3A to the Golgi, suggesting that ORP3A may be involved in the traffic of sterols from the ER to the Golgi[28]. Analogously, in this work we have shown that VAP27-proteins interact with ORP2A, which binds phytosterols. The verified, similar increase in β-sitosterol and campesterol in chloroplasts in all the mutant backgrounds we analyzed (i.e., *vap27-1/vap27-3*, *orp2a-1* and *vap27-1/vap27-3/orp2a*) underscores genetic evidence that ORP2A and VAP27 proteins are functionally linked in maintaining homeostatic levels of sterols in chloroplasts. Interestingly, we found that the *vap27-1/vap27-3*, *orp2a-1*, and *vap27-1/vap27-3/orp2a-1* mutants exhibit an increase in the levels of β-sitosterol and campesterol in chloroplasts. It has been previously shown that the yeast sterol transporter Osh2, a homolog of ORP-proteins, is recruited to ER-endocytic contacts, and that a subset of sterol biosynthetic enzymes also localizes at these ER-PM contacts and interacts with Osh2 and the endocytic machinery. Furthermore, it has been shown that Osh2 extracts sterols from these subdomains, which were named ER sterol exit sites (ERSESs)[88]. Analogously to the yeast ERSESs, which involve the membranes of the ER and PM, the voPLAMs, which encompass the ER and OEM, may be involved in the biosynthesis of sterols and sterol extraction from chloroplasts mediated by ORP2A for transport to other compartments. Therefore, the loss of ORP2A would result in an accumulation of sterols in chloroplasts. Alternatively, ORP2A may function to counterbalance the function of an ER–chloroplast sterol transporter by trafficking sterols away from chloroplasts. Therefore, in conditions of a loss of ORP2A and VAP27-proteins sterols would accumulate in chloroplasts. It cannot be excluded that in addition or in place of sterol trafficking, the VAP27

protein-ORP2A complex may be involved in the actuation or response to processes such signaling or modulating the fluidity of the adjacent ER membrane and OEM. Sterols have been involved in the formation of lipid rafts, which are supposed to play an important role in biological processes, including signal transduction, cytoskeleton reorganization and stress response[89,90]. In this view, a function of the voPLAMs may be to maintain a functional organelle-organelle interface for intracellular signaling and stress response. Indeed, in mammalian cells, the availability of cholesterol has been linked to the regulation of autophagy, a self-digestion process of cellular constituents[91]. It has been recently shown that ORP2A interacts with ATG8e and contributes to autophagy regulation[29]. It is therefore possible that, through an interaction with sterols at the voPLAMS, ORP2A may sense OEM composition and contribute to the control of chlorophagy, a hypothesis that can be tested in the future.

VAPs confer the formation of the ER MCSs with many other membranes but in non-plant species other tethers that include ER-PM tethers (e.g., the calcium-dependent membrane tethers synaptotagmin 1, STIM1/Orai1 channels, anoctamin 8)[92,93], ER-mitochondria tethers (e.g., IP3R/VDAC, Fis1/BAP31)[94,95] and ER-endosome tethers (e.g., Protrudin/Rab7, ORP1L/ORP5)[96,97] are known to exist, underscoring the existence of additional tethers besides VAPs. Similarly, in plant cells the ER membrane-associated synaptotagmin SYT1/SYTA marks ER-PM MCSs, which only partially overlap with the VAP27-associated ER-PM MCSs[98]. Therefore, it is possible that the VAP27–protein–ORP2A complex identified in this work marks a subpopulation of PLAMs. The evidence that a loss of VAP27-1 and VAP27-3 proteins along with ORP2A is viable lends support to this hypothesis and opens the possibility that these proteins have functional redundancy with other sequence homologs and/or other tethers and lipid carriers. Nonetheless, the identification and characterization of VAP27 proteins and ORP2A provided in this work define machinery components of the PLAMs that have been elusive prior to this work and mark a significant step toward future analyses on the physical interaction of the ER with chloroplasts.

## Methods

### Plant materials, growth conditions and isolation of T-DNA knock-out lines
Plants of the *Arabidopsis thaliana* Col0 and Col3 ecotypes T-DNA insertion lines were used in this study were obtained from the Arabidopsis Biological Resource Center (ABRC). The T-DNA insertion lines used in this work are: *vap27-1/vap27-3/orp2a-1*, *vap27-1/vap27-3*, and *orp2a* (see Supplementary Table 1). The triple *vap27-1/vap27-3/orp2a-1* mutant was generated by floral crossing of *vap27-1/vap27-3* double mutant with *orp2a-1*. All the plant-based analyses were performed in mutant homozygous plants, WT background or in stable transformants expressing the indicated constructs. The lines were obtained either through transformation by floral dip followed by antibiotic selection[99] or by crosses with parental marker lines. Arabidopsis seedlings were grown in ½ LS + agar media at 21 °C under a 16 h light/ 8 h dark regime.

### Lovastatin treatment
Arabidopsis seeds for all the lines were germinated respectively in presence of lovastatin at a concentration of 300 nM or the corresponding amount of its solvent as control (DMSO). Plants were grown at 21 °C under a 16 h light/8 for 2 weeks and then their root was measured using Fiji. Experiments were replicated three times; data were statistically analyzed using one-way ANOVA.

### RNA extraction and PCR amplification
RNA was extracted using the RNeasy Plant Mini Kit (Qiagen, http://www.qiagen.com) and either Taq or PFU polymerase was used for the amplification of transcripts. Following mRNA extraction, RT-PCR

amplification of transcripts was carried out using 0.2 mM dNTPs, 0.2 µM primers, and 1 unit of Taq polymerase (Promega, http://www.promega.com), to establish transcript abundance. *Ubiquitin10* (*UBQ10*) was used as reference gene. For cloning, PCR amplification was performed with high-fidelity PFU following the manufacturer's (Biolabs) instructions.

## Plasmids for protein expression in vivo and in vitro, and protein import assays

For in vivo localization analyses *OEP7, VAP27-1, VAP27-3*, and *ORP2A* cDNAs were amplified and subcloned by standard approaches into the binary vector pEARLY-Gate 101 to generate fluorescent protein fusions at the C-terminus of the proteins. For endogenous promoter constructs, the 1000 bp region upstream the start codon was amplified and subcloned upstream either the *VAP27-1-YFP* or *VAP27-3-YFP* sequence. For all the other live-cell imaging analyses, the CaMV35S promoter was used to drive constructs. For the in vitro analyses the pET16b vector was used to express recombinant VAP27-1ΔTMD and VAP27-3ΔTMD and ORP2A as His$_6$ fusions at the proteins N-terminus. VAP27-1 and VAP27-3 were also cloned as GST-fusions at the N-terminus in pGEX5-1 for in vitro protein-protein interaction analyses. For the constructs used in our association assays, VAP27-1ΔTMD and VAP27-3ΔTMD were amplified and cloned into pENTR/SD/DTOPO according to manufacturer protocol (Invitrogen™). Finally, using a LR Clonase II reaction assay as described by the manufacturer protocol (Invitrogen™), VAP27-1ΔTMD-His and VAP27-3ΔTMD-His were cloned into pDEST14. In a similar fashion, Toc33 and AtSSU1B were likewise amplified and cloned into pENTR/SD/DTOPO according to manufacturer protocol (Invitrogen™) and then subsequently cloned into pDEST14 vector. Finally, the luciferase cDNA clone was provided by the TnT® T7 Coupled Wheat Germ Extract System from Promega. Primers were obtained by custom oligonucleotide synthesis (Invitrogen or IDT™). All constructs were confirmed by sequencing. Tobacco leaf infiltration for transient expression in epidermal cells was carried out as described earlier[100].

## Confocal laser scanning microscopy

Confocal image acquisition was performed using an inverted laser scanner confocal microscope Nikon A1Rsi on 10-days-old cotyledon epidermal cells. The fluorescent proteins used in this study were GFP5[101], EYFP, VenusYFP, CFP (Clontech, http://www.clontech.com/). Imaging of the fluorescent markers was achieved as previously described[102–105]. Chloroplast visualization was achieved using chlorophyll autofluorescence excitation at 488 nm excitation and 650–700 nm emission. To perform BiFC experiments the Venus (YFP) was split in two complementary halves, a N-portion of 462 bp and a C-portion of 252 bp, and respectively fused to the protein of interest (see Supplementary Table 1). Tobacco leaves were infiltrated and transiently transformed combining the constructs to be tested for association. ER distribution in non-transformed cells was analyzed using DiOC6 (working solution: 1.8 µM in H$_2$O) (Molecular Probes, http://www.invitrogen.com) for 30 min. For FRAP experiments, fluorochrome photobleaching was performed as described earlier[102] using 3 µm$^2$ areas at the EMPC and 3 µm$^2$ areas at the bulk ER. Number of bleach events computed for the averages = 10 (bulk ER) and 10 (EMPC). Student's two tailed *t* test was used for statistical analysis, assuming equal variance, and data with *P* value ≤ 0.05 were considered significant. Pearson's correlation analyses were used to estimate signal overlap, and one-way ANOVA was used for statistical analysis. The imaging results presented in this work are representative of at least three independent experiments.

## Transmission electron microscopy

Following the immediate harvesting of three-week-old Arabidopsis plant leaves, a series of meticulous steps were undertaken. Leaf sections were immersed overnight in ice-cold 0.1 M cacodylate buffer at a pH of 7.2, enriched with 2.5% (v/v) glutaraldehyde for fixation. Subsequently, they were rinsed in the same buffer, subjected to post-fixation in 1% OsO$_4$ for 2 h at room temperature (RT), and then rinsed once more in cacodylate buffer at pH 7.2. Dehydration was carried out using a gradient of ethanol solutions, and the samples were ultimately embedded in Spurr's epoxy resin, obtained from Electron Microscopy Sciences in Hatfield, PA, USA.

Ultrathin sections, measuring 70 nm in thickness, were sliced using an RMC ultramicrotome from RMC in Tucson, AZ, USA. These sections were mounted onto 150 mesh formvar-coated copper grids, which were also sourced from Electron Microscopy Sciences. In preparation for the analyses, the sections were stained with uranyl acetate for 30 min at room temperature, followed by a thorough rinse with ultrapure H$_2$O. Subsequently, they were stained for an additional 10 min with lead citrate and, once again, rinsed with ultrapure H$_2$O. Finally, images were captured using a JEOL1400Flash instrument from JEOL USA Inc. in Peabody, MA, equipped with a high-sensitivity sCMOS camera. Chloroplast's ultrastructure images were analyzed by Fiji, and data were statistically analyzed by one-way ANOVA with Tukey's post test.

## Protein expression and purification, and in vitro interactions

Recombinant protein production was carried out in *E. coli* BL21. His-ORP2A was extracted using the lysis buffer (50 mM NaH$_2$PO$_4$, 300 mM NaCl, pH 8.0, 1 mM PMSF, and protease inhibitor cocktail 1:200) and then incubated with recombinant GST-VAP27 (ΔTMD) fusions and fixed on resin for GST binding (PrepEase Protein purification Glutathione Agarose 4B USB). Immunoblot analyses were performed using anti-His serum used at a concentration of 1:2000 or GST-serum, using a dilution 1:3000, in both cases an incubation of 2 h was applied[106,107]. Proteins for lipid-binding assay His-VAP27-1, His-VAP27-3 and His-ORP2A were extracted using the lysis buffer (50 mM NaH$_2$PO$_4$, 300 mM NaCl, pH 8.0, 1 mM PMSF and protease inhibitor cocktail 1:200). After extraction proteins were further purified and concentrated using Amicon Filters (10 kDa for the His-VAPs and 50 kDa for His-ORP2A, respectively).

## Lipid extraction and analysis

Total lipids were extracted from 4-week-old *Arabidopsis* leaves with chloroform, methanol, and 88% formic acid (10:20:1 by volume). Next, 0.5 volume of 1 M KCl and 0.2 M H$_3$PO$_4$ was added to the extract and phases were separated by centrifugation (3000 rpm, 3 min, RT). Polar lipids were separated on activated ammonium sulfate-impregnated silica plates (Si250PA; Mallinckrodt Baker, NJ) by using developing solvent of acetone/toluene/water (90:30:7 by volume). Lipids were visualized by staining with iodine vapor and collected by scraping from the plate. Fatty acid methyl esters were separated and identified as described previously[49]. Pentadecanoic acid was used as an internal standard. Data were statistically analyzed by one-way ANOVA with Tukey's post test.

## Sterol extraction and analyses

For sterol extraction from chloroplasts, the chloroplasts were purified from WT Arabidopsis, *vap27-1/vap27-3, orp2A*, and *vap27-1/vap27-3/orp2A* that had been grown on ½ LS, 1% sucrose and 0.8% agar for 3 weeks. Isolated intact chloroplasts were quantified, measuring spectrophotometrically the chlorophyll content for each sample at 663 nm and 645 nm[108]. In total, 500 µL of 1 mg/mL of total chlorophyll equivalent chloroplasts were pipetted into a 13 × 100 mm Pyrex tube with 10 mL Extraction Solvent (for 40 mL extraction solvent: 20 mL of chloroform, 20 mL of methanol, and 0.04 g BHT + 12.5 µg/mL of the internal standard 5α-cholestane and 3β-hydroxy-5α-cholestane previously diluted in chloroform stock to 1 mg/mL in chloroform/methanol (1:1, v/v + 0.1% BHT) in a pre-cleaned 50-mL glass media bottle with

PTFE-lined cap and vortexed thoroughly for few min. 10 ml of 1 M NaCl was then added to the extract and vortexed thoroughly. Phases were separated through centrifugation a 4600×g for 30 min in a tabletop centrifuge at room temperature. The sterol-enriched fraction was then transferred into a fresh 13 × 100 mm Pyrex tube and then evaporated completely dry using nitrogen. The dried sample was then resuspended into hydrolysis solution (for 40 mL 2.0 g of KOH in 28 mL methanol and 12 mL water in a 50 mL polypropylene tube) by vortexing. The tubes were then incubated at 70 °C for 1 h.

After incubation, 1000 µL hexane were added into the tubes and vortexed thoroughly. Samples were then centrifuged again 10 min at 1500×g to separate the two phases, with the sterol-enriched in the top phase that was transferred to a 2 mL autosampler vial and evaporated until completely dry using nitrogen. 100 µL BSTFA (with 1% TMCS) were added to the dry samples, and then vortexed, sealed and incubated at 60 °C for 12–24 h.

For total sterols extraction from leaves, we processed the samples after freezing them in liquid $N_2$. To avoid handling issues for aliquoting ground materials for sterol extraction and protein extraction, we prepared independent batches of samples (10 collections containing ~50 mg of seedlings grown on ½ LS, 1% sucrose and 0.8% agar for 3 weeks, normally 2–3 seedlings) and used 4 batches for sterol, 3 batches for chlorophyll and 3 batches for protein extraction. The procedure was the same as the one used to extract sterols from chloroplasts (described above). Data were normalized by protein content. At the same time, dilution series of the external standards (β-sitosterol, stigmasterol and campesterol) were prepared for each of the sterols for quantification, with a same amount of internal standard (5α-cholestane and 3β-Hydroxy-5α-cholestane) as reference. The sample vials and the standards were transferred into a GC/MS (Gas Chromatography/Mass Spectrometry) autosampler for run and quantification. A list of reagents and their catalogs number is provided in Supplementary material (Supplementary Table 1). Data were statistically analyzed by one-way ANOVA with Tukey's post test.

Data were analyzed by multiplying the concentration by the total volume of derivatization reagent to get the total amount of sterols in the vial. Normalization was done by dividing this number by the chlorophyll content (i.e., 1 mL of 1 mg/mL chlorophyll).

## Fat blot assay and sterol-binding assay

Fat blot spotting assays were performed according to ref. 109. In total, 15 µg of either phytosterols (β-sitosterol, stigmasterol, and campesterol) or 15 µg of OEM galactolipids (MGDG, DGDG, PG) and chloroform (solvent control) were spotted on nitrocellulose membrane (Amersham Hybond-C Extra -RPN203E) and let to dry. The controls PI and PI3P at a concentration of 100 pM (VAP27 lipid-binding assay) and 200 pM (Orp2A lipid-binding assay) were used. The membranes were successively blocked with TBS-T + BSA for 2 h then handled as described in the protocol for Lipid Strip and Lipid Array Products by Echelon Biosciences Inc. website. The nitrocellulose membrane was then incubated overnight with the purified His-tagged proteins (~15 h). After washing three times for 10 min in TBS-T, the membrane was incubated with anti-HIS serum (Santa Cruz biotechnology cat#sc-804) diluted 1:2000 for 3 h. After this step, the washes were applied to the membranes as indicated above. The secondary antibody was incubated for 1 h, and the membrane was then developed using ECL solution after the third round of washes. The same procedure was followed for sterols binding assay; in this case the sterols that were subjected to binding were spotted on the same kind of membrane used for lipids. The rest of the procedure was identical to the one described for lipids above.

## Chloroplasts preparation

For protein−chloroplast association analyses, intact chloroplasts were isolated from either 8- to 12-day-old pea seedlings (Little Marvel (Dwarf Variety), Urban Farmer, Westfield, IN 46074) or 2-3 week-old plate grown Arabidopsis Columbia plants and purified over a Percoll gradient[110]. Intact pea or Arabidopsis chloroplasts were reisolated and resuspended in import buffer (330 mM sorbitol, 50 mM HEPES/KOH, pH 8.0) at a concentration of 1 mg chlorophyll mL$^{-1}$ [111].

## Immunoblotting on isolated Arabidopsis chloroplasts

Total Arabidopsis chloroplast membranes (5 µg chlorophyll/lane) isolated from various backgrounds were resolved by 4-20% SDS-PAGE (Bio-Rad™ precast gel), and proteins were transferred to Immobilon-P (Millipore-IPVH00010) using a Bio-Rad tank transfer system. Proteins were transferred to Immobilon-P for 1 h and 100 V. After transfer, the Immobilon-P blot was soaked in 5%(w/v) dry Milk (DM) in 20 mMTris, 150 mM NaCl, 0.1%Tween 20 (TBST) for 30 min, with constant rocking. Blots were then washed 2× with TBST (approximately 30 s/each). Blots were then incubated with various primary antibodies in 5%DM/TBST overnight with slow rocking at 4 °C. After overnight incubation, blots were then washed 3× with TBST for 5 min each with vigorous shaking. Blots were incubated with a secondary antibody (goat anti-rabbit IgG-alkaline phosphatase conjugate (ThermoFisher G-21079) in 5%DM/TBST for 1 h with slow rocking at room temperature. Blots were then washed 3X with TBST for 5 min each with vigorous shaking. Finally, blots were developed using incubation in ThermoScientific 1-Step™ NBT (nitro-blue tetrazolium chloride)/BCIP (5-bromo-4-chloro-3'-indolyphosphate p-toluidine) solution (Cat# 34042) for 5–15 min or until color developed. The reaction was quenched by rinsing blots with water.

## Antibodies

The antibodies used in this study were the following: anti-Toc75 (N-terminal region; 1:2000 dilution), anti-Toc33 (PHYTO A/B PHY0650A, 1:1000 dilution); anti-Tic110 (from Keegstra Lab; 1:2000 dilution), anti-LHCB1/3, N-terminal (PHYTO A/B PHY0085A, 1:4000 dilution); anti-SMT1 (Agrisera AS07 266; 1:500 dilution), anti-CESA1 (PHYTO A/B, PHY0798S, 1:1000 dilution), anti-His (Santa Cruz SC-8036), and anti-GST (Santa Cruz SC-138) using respectively 1:2000 and 1:3000 diluitions.

## In vitro translation of precursor proteins and protein association assays

All proteins used in our association assays were radiolabeled using [³H]-Leucine (approximately 0.05 mCi/50ul Translation Reaction) and translated using Promega's TNT® Coupled Wheat-germ Lysate System according to the manufacturer's protocol. After translation, [³H]-labeled proteins were diluted with an equal volume of "cold" 50 mM L-Leucine in 2× Import Buffer (IB) (Froehlich 2011). For the protein association assays, 100 µl chloroplasts (1 mg chlorophyll/ml), 4 mM Mg-ATP/IB final concentration, and 100 µl radiolabeled proteins (i.e., TP: translation product; either ³H-ORP2A, ³H-Luciferase, ³H-Toc33 or ³H-AtSSU1B) were added to a final volume of 300 µl, and incubated for 30 min at room temperature, under room light. Upon completion of the association assay, the reaction was divided into two 150 µl aliquots. One portion was not further treated with protease [(-) control] and intact chloroplasts were directly recovered by centrifugation through a 40% Percoll cushion. The other portion was incubated with trypsin for 20 min on ice. After quenching trypsin with trypsin Inhibitor, intact chloroplasts were again recovered by centrifugation through a 40% Percoll cushion. All total chloroplast fractions were subsequently analyzed using SDS-PAGE. After electrophoresis, the SDS-PAGE gel was subjected to fluorography and exposed to X-ray film (Eastman, Kodak, Rochester, NT, USA).

## Reporting summary

Further information on research design is available in the Nature Portfolio Reporting Summary linked to this article.

## Data availability

The authors declare that the data supporting the findings of this study are available within the Article and its Supplementary files, as detailed in the Reporting Summary. Other data supporting the conclusions of the study are available from the corresponding author. Source data are provided with this paper.

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

## Acknowledgements

The authors thank Dr. Agnieszka Zienkiewicz, MSU-DOE Plant Research Lab, for starting the lipidomics analyses and the Keegstra lab, MSU-DOE Plant Research Lab, East Lansing for antibodies to Toc75 and Tic110. The authors thank the Mass Spectrometry & Metabolomics Core Facility, particularly Dr. Johnny Casey, for helping with protocol setup and quantification. This work was funded primarily by the Chemical Sciences, Geosciences and Biosciences Division, Office of Basic Energy Sciences, Office of Science, US Department of Energy (award number DE-FG02-91ER20021) with contributing support from the Great Lakes Bioenergy Research Center, U.S. Department of Energy, Office of Science, Office of Biological and Environmental Research (DE-SC0018409) and AgBioResearch (MICL02598) to FB. Anastasiya Lavell was partially supported by a fellowship from Michigan State University under the Training Program in Plant Biotechnology for Health and Sustainability (T32-GM110523).

## Author contributions

L.R., G.S., and F.B. designed the research. L.R., G.S., C.B., and F.B. interpreted the results. L.R., G.S., M.P.P., S.J.K., J.E.F., and G.B. performed the experiments. L.R. and F.B. wrote the paper. C.B. and S.M. helped with resources and infrastructures.

## Competing interests

The authors declare no competing interests.
