## [Peer Review File · Nature Communications]

Reviewers' Comments:

Reviewer #1:

Remarks to the Author:

Membrane contact sites have been extensively studied in the past decade in non-plant systems. Though the interaction between ER and chloroplast has been reported by previous studies, information on the nature of endogenous PLAMs is still missing. Renna et al. reported VAP27-ORP2A complex defines the ER-chloroplasts contact and bridges the lipid transport between ER and chloroplast. The manuscript was well prepared. However, the VAP27-ORP2A complex and its roles in lipid transport have been reported recently and the message is not so new. And In addition, I found several problems as below.

1. The relationship between ER and chloroplasts shown in Fig 1 and Fig 2 is detected using 2D imaging. The ER overlapped with chloroplast from the image, but whether the interactions between the two organelles occur at this site still remains unknown.
2. According to previous studies, the mobility of proteins on tubule/bulk flow/cisternae ER are rather distinct from each other. The mobility of VAP27 proteins in the EMPC should be detected at the same type of ER membrane.
3. In the BiFC results, there are many fluorescent puncta localized between the chloroplasts. However, the OEP has been reported to be an OEM marker. Why it can be detected to interact with VAP27 in the cytosol? Moreover, the positive control and negative control should be included in the figure or the supplemental figure.
4. The authors tested the lipid composition using the leaf tissue to study the function of VAP27 and ORP2A. Why did not use the isolated chloroplasts?
5. Several tether proteins have been reported in membrane contact sites in plants, for example, SYT1 is also widely involved in ER-related membrane contact sites. Indeed, the VAP27 accumulate at ER-chloroplasts in this study. Does SYT1 or any other tether proteins also participate to form the complex? From the results obtained in this manuscript, the authors can not conclude that "The plant ER connects with chloroplasts at regions where VAP27 proteins accumulate".
6. The sterol homeostasis was influenced in the *orp2a-1*, *vap27-1/vap27-3*, and *vap27-1/vap27-3/orp2a-1*. Is the ER-chloroplasts interaction also affected in these mutants? Moreover, the mesophyll cell size and chlorophyll content were dramatically changed in *orp2a-1* mutants, does the phenotype can also be detected in the *vap27-1/vap27-3* and *vap27-1/vap27-3/orp2a-1*?
7. The scale bar missed in the supplemental figure 2D
8. All the images showing overlapping should be statistically analyzed.
9. The references should be updated, there are many beautiful works in the MCSs published in the past five years.

Reviewer #2:

Remarks to the Author:

This paper tries to identify protein involved at chloroplast-ER contact sites. The authors started with VAP27-1 and VAP27-3 as candidates and found they are localised in the ER as previously described but concentrate also in close association to the chloroplast. They then realized yeast double hybrid screen and identify ORP2A as an interactor of both VAP27. To address the localisation of this protein regarding the chloroplast they carried out protein association assay and found these three proteins are associated to the OEM. Finally, the authors purified *orp2a-1* KO mutant, used their previously published double mutant *vap27-1/vap27-3* and constructed a triple mutant. They analyse the distribution of the leave glycerolipids as well their fatty acid distribution but not their quantification, and they analysed the plastid sterol composition where they found some differences between the mutants and the WT. As a conclusion, they proposed that the VAP27-ORP2A complex defines a ER-chloroplast MCS involved in chloroplast sterol metabolism This paper is well written. The work is nicely done and the experiments are well described. However, this paper comes just after the publication of a PNAS paper that establishes the interaction between VAP27-1 and ORP2A in ER-autophagosomal contact sites (DOI:

10.1073/pnas.2205314119). The interaction between VAP27 and ORP2A is therefore not new. The focus of this article is different, concentrating on plastid and sterol inside the chloroplast. However, the sterol content of chloroplast is not very well documented in the literature and still debated. Here the chloroplast sterol content is expressed in ng/ μ L (that I guess is correlated to a standardize chlorophyll amount) but do not give an idea of the real abundance of the sterol and no data for the total leaf content is shown whereas the opposite is done for the glycerolipid profile. For instance, the plasma membrane is the membrane that contain the highest amount of sterol and the contamination by plasma membrane of the chloroplast fraction is not evaluated... This is the main result of this paper that is different of what was shown in previous literature and it raises question.

Major remark:

- What is the phenotype of the mutants? in the Figure S2, the mutant *orp2a-1* looks sick. If there is an impairment of autophagy, it is not surprising. Is there in microscopy an impact on the chloroplast structure? Why just show the root phenotype (root length) and not the leaves phenotype when the focus is on chloroplast? It would have been interesting to have the lipid content of the mutants and not only the composition. Why do not look at the chloroplast lipid composition as it is done for the sterol?
- For the sterol analysis, what is the global profile and content of the leaves? Is there an effect of the mutants in the global sterol content? If yes, is it similar to what is happening in the chloroplast? Is the chloroplast fraction devoid of plasma membrane contamination?
- In the PNAS paper, ORP2A is binding phosphoinositides as well as VAP27. Is there any phosphoinositides in the chloroplast? Are the chloroplast bound to VAP27 and ORP2A in an autophagy process?

Minor remark

- Sterol content is express in ng/ μ L. By reading the material and method, I suspect it is ng/ μ g of chlorophyll. How does it compare to literature data? Can it be expressed in ng /mg of protein and compare with what is known for the plasma membrane?
- The usage of VAP27 and VAP27delta TM is not always clear... apart from the yeast two hybrid, why use the truncated version?

Reviewer #3:

Remarks to the Author:

Authors have found out that VAP27-1 and VAP27-3 (endoplasmic reticulum proteins) concentrate in areas surrounding chloroplast and that these proteins interact with ORP2-A (a putative lipid transfer protein). Additionally, they describe that this association is involved in maintaining sterol homeostasis between ER and chloroplast. This work is original, it is well described and clear. All in all, this study is well performed and of interest for the plant membrane contact site and plant lipids communities. However, I think that conclusions about subcellular localization, interaction and function of these proteins are stronger than the evidence that has been shown.

Comments:

Title: In this work, it has been proven the association of VAP27-1 and VAP27-3 with ORP2A (there is no evidence for the rest of VAP proteins). Additionally, from my point of view, there is not strong evidence to claim to be a "functional" and "complex".

Introduction: It contains all the needed information with the right references.

Results

- Section 1: The ER membrane proteins VAP27-1 an VAP27-3...)

Authors are showing one single chloroplast and one single Pixel intensity plot to demonstrate that these ER proteins are concentrated in areas of chloroplast association. As these are transient expression experiments, I wonder if this could also be performed for rest of VAP proteins. I also think the manuscript could be improved if they include less zoomed images (showing a bigger part of the cell), to be able of detect how frequent are these associations. Additionally, the authors could provide some quantitative analysis on these analyses of pixel intensity to reach out a robust and unbiased conclusion.

It is also worth noting that the expression pattern of these proteins when using their own promoters is very different from the transient expression experiments or the Arabidopsis lines using the 35S promoter.

Lines 156 to 157: Authors have analyzed VAP27-1 and VAP27-3 subcellular localization in Tobacco and Arabidopsis. They have found out that the stable expression with their own promoters is different when they used constitutive promoters (lines 150-155). I do not think it has been proven that the subcellular localization of all VAP27 proteins is conserved inter-species.

A feature of BiFC is the irreversibility of the FP reconstitution. And it frequently stabilizes self-assemble FP partners at high local concentrations. Thus, a negative control is important to draw a clear conclusion in this experiment.

- Section 2: VAP27-1 and VAP27-2 interact with ORP2A...

Authors have proven the interaction of these proteins by Y2H and in vitro pulldown experiments. But they have already generated Arabidopsis stable lines expressing VAP27-CFP and ORP2A-YFP. I think they could provide stronger evidence for this interaction by performing Co-IP of these two proteins in Arabidopsis. Then, they will have evidence of the in vivo interaction. Additionally, to get better evidence of this interaction in vivo and at the PLAMS they could perform FRET-Flim experiments. A better negative control for the in vitro pulldown assays would also be good. For example, could you perform similar experiments with truncated version of these proteins? (It will also help in the identification of the interaction area). Or with any of the other VAP27 proteins (for a negative interaction?) Do all VAP27 proteins interact with ORP2A? This is important to know when discussing sterol analysis (below).

Figure 4C: I would suggest the authors to show a bigger area of the cells. Are they only colocalizing at this subcellular area? Have they performed any quantification?

-Section 3: ORP2A and VAP27 proteins associate with the OEM

Authors have performed several experiments to gain insight into the association of these proteins with the OEM by using Trysin. These experiments have clearly demonstrated that the interaction of ORP2A with OEM. However, their studies about the binding of ORP2A and VAP27 to lipids is poor. I would suggest using a bigger collection of lipids (including PI, PC, PS, PE, DAG, TAG, sterols...)

Line 255: I do not think it has been proven these proteins form a complex.

Line 256: I do not think it is proven these proteins are binding "OEM signature glycerolipids".

- Section 4: The loss of VAP27 and ORP2A...

I agree with the authors, changes in glycerolipids are very subtle.

- Section 5: ORP2A is necessary for sterol homeostasis

Authors have analyzed major sterols in WT, *orp2a-1*, *vap27-1/vap27-3*, and *vap27-1/vap27-3/orp2a-1* mutants. And they interestingly have found out that mutant plants accumulate more sterols (in particular sitosterol and campesterol) than WT plants. However, it is surprising that *vap27-1/vap27-3* is accumulating sterols at the same levels as the *orp2a-1* and the triple mutant. These could point to *vap27-1* and *-3* being the two only VAP27 proteins at the PLAMS. I would

suggest to better discuss this in the discussion.

I also think it could be interesting to show a picture of these plants (in the supplementary part).

Are they showing the same phenotype?

Finally, as chloroplast are not the main site for sterol accumulation, it would be good if the authors can show the sterol composition in all total tissue like leaves (not just in chloroplast). Did they notice any changes?

Discussion:

Line 338: I do not think it is completely right to assert that VAP27-1 and VAP27-3 proteins preferentially accumulate at PLAMs.

Lines 336 and 337: I do not think it is proven. I think you should add other lipids to the analysis so they can be negative controls.

Lines 401-403: It is really difficult to understand how VAP27 proteins, that are known to be localized at other MCSs , are binding MGDG. I think authors should discuss this.

We would like to thank the reviewers for their insightful comments. We have addressed all their suggestions and comments by including new results and editing the manuscript.

We believe that the reviewers' points were constructive and helped us improve our work significantly.

Our detailed responses are embedded in the text below.

REVIEWER COMMENTS

Reviewer #1 (Remarks to the Author):

Membrane contact sites have been extensively studied in the past decade in non-plant systems. Though the interaction between ER and chloroplast has been reported by previous studies, information on the nature of endogenous PLAMs is still missing. Renna et al. reported VAP27-ORP2A complex defines the ER-chloroplasts contact and bridges the lipid transport between ER and chloroplast. The manuscript was well prepared.

However, the VAP27-ORP2A complex and its roles in lipid transport have been reported recently and the message is not so new. And In addition, I found several problems as below.

- Thank you for the comment. This manuscript is the fruit of many years of work and collaboration with a major expert in the field of lipid trafficking. We respectfully disagree with the reviewer that the role of the VAP27-ORP2A complex in lipid transport has been reported recently. The role of VAP27-ORP2A interaction in lipid transport has not been described in plants. A binding of ORP2A to lipids has been shown by the Liwen Jiang's group, but no functional insights on lipid transport per se have been proposed in their work (doi.org/10.1073/pnas.2205314119). Furthermore, in that work the interaction of VAP27 with ORP2A has been exclusively studied in the context of ER-plasma membrane contacts sites. Therefore, our data showing that the VAP27-ORP2A complex is localized also at the ER-chloroplast contact sites are novel. We additionally provide a functional and molecular characterization of these sites, which, as also acknowledged by the reviewer, is novel.

The relationship between ER and chloroplasts shown in Fig 1 and Fig 2 is detected using 2D imaging. The ER overlapped with chloroplast from the image, but whether the interactions between the two organelles occur at this site still remains unknown.

- Our results are based on published biochemical and ultrastructural analyses showing that the ER and chloroplasts physically interact with each other (doi.org/10.1074/jbc.M608124200). Those published studies are also supported by functional analyses that the ER and chloroplast interact to produce essential lipids in the cell (doi.org/10.1021/pr034025j). Our 2D imaging results are in alignment with these

earlier findings. A physical contact of the ER with the chloroplasts at the sites described in our manuscript is provided by the biomolecular fluorescence complementation analyses (BiFC), which are based on protein-protein interactions on closed proximity. The BiFC experiments show reconstitution of YFP at the EMPC marked by VAP27 proteins (Figure 3). In the revised manuscript, we have integrated our 2D results with Z-Stacks maximum projection imaging datasets and quantification. The results confirm two sets of original evidence that the ER marked by VAP27 interacts with chloroplasts at the EMPC.

According to previous studies, the mobility of proteins on tubule/bulk flow/cisternae ER are rather distinct from each other. The mobility of VAP27 proteins in the EMPC should be detected at the same type of ER membrane.

- We agree with the reviewer that the mobility of proteins on tubule/bulk flow/cisternae ER are distinct from each other. Proteins have indeed specific dynamics in different domains of the ER. For instance, the mobility of VAP27 proteins is restricted at the ER-PM contact sites compared to the bulk ER (doi.org/10.1016/j.cub.2014.05.003). Therefore, to address the comment of the reviewer, we measured VAP27 mobility at the level of ER tubules and the EMPC, using fluorescence recovery after photobleaching (FRAP). As a control in the analyses, we used a bulk ER membrane marker based on a fluorescent protein fusion to the transmembrane domain and cytosolic tail of Calnexin. This is an inert marker of the ER membrane (bulk ER) (doi.org/10.1093/jxb/erg102). This experiment served as a control to establish how VAP27 mobility differed from that of a protein that does not localize preferentially at the EMPC. We added these new results in Figure 2B.

3. In the BiFC results, there are many fluorescent puncta localized between the chloroplasts. However, the OEP has been reported to be an OEM marker. Why it can be detected to interact with VAP27 in the cytosol? Moreover, the positive control and negative control should be included in the figure or the supplemental figure.

- VAP27 does not localize in the cytosol, therefore the puncta localization are unlikely cytosolic. Because we performed all our confocal acquisition using pinhole 1, which provides a very narrow focal plane, the puncta are most likely regions of the ER that connect with plastids but where the bulk ER is not visible because outside of the focal plane of the acquisition. We agree with the reviewer's request to include a control. So, we added a positive a negative control using OEP7 and the ER membrane protein RHD3 (doi.org/10.1242/jcs.084624; doi.org/10.1111/j.1365-313X.2011.04846.x), fused to a complementary half YFP. RHD3 is a membrane protein localizing to the bulk ER. The results have been added as a supplemental figure (Fig. S2A).

The authors tested the lipid composition using the leaf tissue to study the function of VAP27 and ORP2A. Why did not use the isolated chloroplasts?

- Chloroplast lipids are quite distinct. For example, the glycolipids MGDG and SQDG only occur in plastids, DGDG is not found outside of plastids unless the plant is severely under phosphate stress – which is not the condition adopted in this work -, and PG has molecular species that contain 16:1. Hence a leaf extract provides a reasonable complete picture of the major chloroplast glycerolipids and how they might be affected in mutants. For the lipid composition analyses, we followed well established procedures to study chloroplast lipid mutants.

Several tether proteins have been reported in membrane contact sites in plants, for example, SYT1 is also widely involved in ER-related membrane contact sites. Indeed, the VAP27 accumulate at ER-chloroplasts in this study.

Does SYT1 or any other tether proteins also participate to form the complex? From the results obtained in this manuscript, the authors can not conclude that “The plant ER connects with chloroplasts at regions where VAP27 proteins accumulate”.

- Our lab and others have carried out protein-protein interaction assays using VAP27 proteins as baits but have not identified SYTA/1 proteins as potential interactors (e.g., doi.org/10.1016/j.celrep.2018.04.091). This is plausible considering that ER-plasma membrane contact sites labelled by either VAP27 or SYTA/1 proteins do not completely overlap (doi.org/10.1093/jxb/erw381), indicating that the two protein families may not have overlapping roles and therefore a physical interaction. This is also supported by the evidence that the loss-of-function phenotypes of VAP27 proteins and SYT proteins are not similar (doi.org/10.1093/jxb/erw381). Therefore, it cannot be expected that VAP27 proteins co-localize with SYT proteins at the EMPC. Nonetheless, to address the point of the reviewer, we tested the subcellular localization of SYTA-YFP in tobacco leaf epidermal cells. We found no preferential distribution of this protein at the EMPC. The results have been added to a supplemental figure (Fig. S1A). Concerning the point of the reviewer that the authors “can not conclude that the plant ER connects with chloroplasts at regions where VAP27 proteins accumulate”, we interpret that the reviewer suggested that our data do not allow us to establish that membrane of the ER is in contact with the outer envelope protein. As detailed in a previous comment to the reviewer, physical contact of the ER with the chloroplasts at the sites described in our manuscript is provided by the BiFC analyses, which are based on protein-protein interactions grounded on closed proximity. The BiFC experiments show reconstitution of YFP at the EMPC marked by VAP27 proteins (Figure 3). Such a direct contact was shown by the Sandelius group (doi.org/10.1074/jbc.M608124200) when they described the PLAMs for the first time. Biochemical evidence proposed by those authors indicated that the PLAMs are membranes with chloroplast and ER constituents. To address the reviewer’s comment in the revised text, we modified our statement to read “VAP27 accumulate in the regions where ER is juxtaposed with chloroplasts” because we have observed a physical overlap of these regions marked by VAP27 with the chloroplasts, which we then assayed with BiFC for a close association. Furthermore, we have edited a paragraph heading to read:

“The plant ER connects with chloroplasts at regions where VAP27 proteins exhibit reduced mobility and interact with the OEM”.

The sterol homeostasis was influenced in the *orp2a-1*, *vap27-1/vap27-3*, and *vap27-1/vap27-3/orp2a-1*. Is the ER-chloroplasts interaction also affected in these mutants?

- To address the reviewer’s point, we used DiOC6, a lipophilic marker of the ER (doi.org/10.1104/pp.17.01261). This allowed us to analyze the morphology of the ER at the interface with the chloroplasts to understand if the ER shapes differently around this organelle in the *orp2a-1*, *vap27-1/vap27-3*, and *vap27-1/vap27-3/orp2a-1* mutants compared to WT. We quantified the levels of overlap between the ER membranes and the chlorophyll autofluorescence and found that the loss of VAP27 proteins and ORP2A in the respective double, single and triple mutant combinations resulted in a small but similarly reduced correlation coefficient compared to WT. The data have been included in a supplemental figure (Fig. S4). The results indicate that in the absence of ORP2A and VAP27 the association of the ER and chloroplast is compromised, which supports our sterol quantification results. Interestingly, also while the loss of VAP27 and ORP2A alters sterol homeostasis, the triple *vap27-1/vap27-3/orp2a-1* mutant is viable and shows some levels of growth reduction. Because sterol deficient mutants exert strong growth reduction (doi.org/10.1104/pp.014605), our results indicate that proteins we identified at ER-chloroplast contact sites function in a manner that is partially redundant with other proteins the ER-chloroplast contact sites. This hypothesis is presented in the discussion.

Moreover, the mesophyll cell size and chlorophyll content were dramatically changed in *orp2a-1* mutants, does the phenotype can also be detected in the *vap27-1/vap27-3* and *vap27-1/vap27-3/orp2a-1*?

- Good point. We revised the supplemental figure where we reported a graph representing the mesophyll size in all the mutants suggested by the reviewer as indication of the growth phenotype observed in all the lines. The data have been included in a supplemental figure (Fig. S3C).

7. The scale bar missed in the supplemental figure 2D

- We added the scale bar. Thank you.

8. All the images showing overlapping should be statistically analyzed.

- Done. We performed a Pearson’s correlation coefficient (doi.org/10.1002/cyto.a.20896) to quantify our colocalization results and provide a statistical evaluation.

9. The references should be updated, there are many beautiful works in the MCSs published in the past five years.

- Among the very interesting and inspiring works that lately have been published on MCSs, we tried focusing on literature related to the PLAMS. Nonetheless, we included more updated literature on the MCSs in the revised introduction. Thank you.

Reviewer #2 (Remarks to the Author):

This paper tries to identify protein involved at chloroplast-ER contact sites. The authors started with VAP27-1 and VAP27-3 as candidates and found they are localised in the ER as previously described but concentrate also in close association to the chloroplast. They then realized yeast double hybrid screen and identify ORP2A as an interactor of both VAP27. To address the localisation of this protein regarding the chloroplast they carried out protein association assay and found these three proteins are associated to the OEM. Finally, the authors purified orp2a-1 KO mutant, used their previously published double mutant vap27-1/vap27-3 and constructed a triple mutant. They analyse the distribution of the leave glycerolipids as well their fatty acid distribution but not their quantification, and they analysed the plastid sterol composition where they found some differences between the mutants and the WT. As a conclusion, they proposed that the VAP27-ORP2A complex defines a ER-chloroplast MCS involved in chloroplast sterol metabolism

This paper is well written. The work is nicely done, and the experiments are well described. However, this paper comes just after the publication of a PNAS paper that establishes the interaction between VAP27-1 and ORP2A in ER- autophagosomal contact sites (DOI: 10.1073/pnas.2205314119). The interaction between VAP27 and ORP2A is therefore not new.

The focus of this article is different, concentrating on plastid and sterol inside the chloroplast. However, the sterol content of chloroplast is not very well documented in the literature and still debated.

- Thank you for the positive comments. We agree with the reviewer that there are only very few reports focusing on sterol quantification in chloroplasts. Sterols have been mostly quantified from a total leaf extract because the scientific interest has been so far to evaluate phytosterol content at general biosynthetic level and general impact on plant physiology. In our case, starting from known sterols identified in chloroplasts we wanted to evaluate the possibility of a different sterol content between WT, VAP27 and ORP2A mutant backgrounds due to a defect in their transport from ER to chloroplast. Therefore, we proceeded with a sterol quantification specifically in chloroplasts. We additionally tested a possible interaction of the oxysterol binding protein ORP2A. Therefore, as correctly stated by the reviewer, although the interaction of VAP27 and ORP2A is known

(doi.org/10.1073/pnas.2205314119), the binding of ORP2A to sterols presented in our work is novel. In the publication from the Liang group, building upon the known role of VAP27 in autophagy at the ER-plasma membrane contact sites (doi.org/10.1038/s41467-019-12782-6), the focus of the work was about the role of ORP2A in autophagy at these sites. Therefore, the establishment of a role of ORP2A in sterol transport at the ER-chloroplasts contact sites is novel and, tangibly, is a different story from that from the Liang's group. Above all, the identification of the machinery functionally interfacing the ER-chloroplasts is novel as well. Most likely, we did not explain these concepts well in the original manuscript and we have revised our manuscript accordingly.

Here the chloroplast sterol content is expressed in ng/ μ L (that I guess is correlated to a standardize chlorophyll amount) but do not give an idea of the real abundance of the sterol and no data for the total leaf content is shown whereas the opposite is done for the glycerolipid profile. For instance, the plasma membrane is the membrane that contain the highest amount of sterol and the contamination by plasma membrane of the chloroplast fraction is not evaluated... This is the main result of this paper that is different of what was shown in previous literature, and it raises question.

- Thank you for your insightful comment. We focused on chloroplasts because we wanted to quantify the sterols specifically synthesized and shuttled between the ER and the chloroplasts. We already mentioned above specific signature glycolipids for chloroplasts, such as MGDG or SQDG, or DGDG under normal growth conditions and specific species of PG, all of which are much more abundant than sterols and can be analyzed differently and with simpler protocols. However, to our knowledge there are no reports of specific signature sterols in chloroplasts. Furthermore, because of the sterols' much lower abundance and different chemical behavior, it was necessary to isolate and purify chloroplasts, and analyze their sterol content by GC-MS. This approach allowed us to pinpoint differences in sterol content in the chloroplasts of WT compared to the mutant that would have been likely missed by doing whole leaf extracts. The reviewer is correct that sterols accumulate preferentially in the plasma membrane after their synthesis at the ER. In our original work, we tested the chloroplast extracts for bulk ER contamination and found none. In the revised manuscript, we tested for PM contamination using antibodies for the plasma membrane multi-spanning protein cellulose synthase (CESA1). We did not find any PM contamination. These results allow us to conclude that our chloroplast isolation was successful, and that plasma membrane contamination does not influence the sterol content in the isolated chloroplasts. We thank the reviewer for their insightful comment, as it allowed us to consolidate our data. The CESA1 control has been added to a supplementary figure (Figures 7C, S7B). As for the sterol content we had to normalize to the chloroplast content which is, as the reviewer well guessed, referred to chlorophylls content. The chloroplast sterol content is expressed in ng/ μ L which is relative to 1 mg/ml chlorophyll equivalent of chloroplasts, all the data have been properly normalized.

Major remark:

- What is the phenotype of the mutants? in the Figure S2, the mutant *orp2a-1* looks sick. If there is an impairment of autophagy, it is not surprising. Is there in microscopy an impact on the chloroplast structure?

- The reviewer raises an excellent point. We included ultrastructure analyses to monitor changes in the thylakoid ultrastructure and established different thylakoids morphology in the mutants. The results are in line with the phenotype of an altered lipid composition effect in thylakoids (doi.org/10.1007/s10265-016-0827-y). The results have been included in a supplementary figure (Fig. S5).

Why just show the root phenotype (root length) and not the leaves phenotype when the focus is on chloroplast? It would have been interesting to have the lipid content of the mutants and not only the composition. Why do not look at the chloroplast lipid composition as it is done for the sterol?

- For the sterol analysis, what is the global profile and content of the leaves? Is there an effect of the mutants in the global sterol content? If yes, is it similar to what is happening in the chloroplast? Is the chloroplast fraction devoid of plasma membrane contamination?

- In the PNAS paper, ORP2A is binding phosphoinositides as well as VAP27. Is there any phosphoinositides in the chloroplast? Are the chloroplast bound to VAP27 and ORP2A in an autophagy process?

Minor remark

- Sterol content is express in ng/ μ L. By reading the material and method, I suspect it is ng/ μ g of chlorophyll. How does it compare to literature data? Can it be expressed in ng /mg of protein and compare with what is known for the plasma membrane?

- We analyzed the root phenotype as a hallmark of organ growth, which can be hampered by a variety of situations, including defects of the function of the aerial organs. Following the root growth phenotype allowed us to report carefully on the impact of lovastatin, an inhibitor of the mevalonate (MVA) pathway (doi.org/10.1093/pcp/pcm005), which is important in sterol homeostasis. We already discussed above that the bulk glycerolipids can be readily analyzed in leaf lipid extracts, while chloroplast sterols cannot because of their lower abundance and because no chloroplast specific sterols have been described. Moreover, because the focus here is on exchange of sterols with the ER, the major aim of our experiment is to measure sterols in a highly purified chloroplast fraction and compare the various lines to establish how the absence of VAP27 and ORP2A affects the sterol content in chloroplasts. While it is possible that the loss of VAP27 and ORP2A alters the total levels of sterols, the evidence that VAP27 and ORP2A interact at the ER-chloroplast contact sites and that their loss alters the sterol content of chloroplasts argues that these proteins are necessary for the homeostasis of sterols in chloroplasts.

- The usage of VAP27 and VAP27delta TM is not always clear... apart from the yeast two hybrid, why use the truncated version?

- We used the terminology VAP27ΔTMD to indicate truncation of the transmembrane domain (TMD) of VAP27. This is necessary for obtaining a soluble protein that can be used in yeast two hybrid analyses as well as for *in vitro* protein-protein interaction and *in vitro* association assays. Removing the TMD is a necessary step to increase protein solubility for expression in E.coli BL21 cells. Other than in these two cases, the full-length proteins are used in all the other experiments. We clarified this point in the revised manuscript. Thank you.

Reviewer #3 (Remarks to the Author):

Authors have found out that VAP27-1 and VAP27-3 (endoplasmic reticulum proteins) concentrate in areas surrounding chloroplast and that these proteins interact with ORP2-A (a putative lipid transfer protein). Additionally, they describe that this association is involved in maintaining sterol homeostasis between ER and chloroplast. This work is original, it is well described and clear. All in all, this study is well performed and of interest for the plant membrane contact site and plant lipids communities. However, I think that conclusions about subcellular localization, interaction and function of these proteins are stronger than the evidence that has been shown.

Comments:

Title: In this work, it has been proven the association of VAP27-1 and VAP27-3 with ORP2A (there is no evidence for the rest of VAP proteins). Additionally, from my point of view, there is not strong evidence to claim to be a “functional” and “complex”.

- As the reviewer suggested we better specified the VAP27 identity in the title. We did so in the revised manuscript. Thank you.
- The genetic results coupled with the biochemical and phenotypical analyses provided in the manuscript show that VAP27 and ORP2A form a complex and that such complex is functional. This is supported by the genetic evidence that the loss of VAP27 and ORP2A proteins alters the levels of sterols in chloroplasts and that the high-order mutant of VAP27 and ORP2A combined shows a sterol phenotype similar to the single mutant.

Introduction: It contains all the needed information with the right references.

- Thank you for carefully checking this.

Results

- Section 1: The ER membrane proteins VAP27-1 and VAP27-3...

Authors are showing one single chloroplast and one single Pixel intensity plot to demonstrate

that these ER proteins are concentrated in areas of chloroplast association. As these are transient expression experiments, I wonder if this could also be performed for rest of VAP proteins. I also think the manuscript could be improved if they include less zoomed images (showing a bigger part of the cell), to be able of detect how frequent are these associations. Additionally, the authors could provide some quantitative analysis on these analyses of pixel intensity to reach out a robust and unbiased conclusion.

- We focused on chloroplasts because we aimed to show the region where VAP27 and ORP2A proteins accumulate. We would like to specify that the intensity plot is not done on 1 pixel but on an entire region of interest (ROI). We better explained this in the methods and results sections and provided the ROI size using pixel as unit. As requested by the reviewer, in the revised manuscript we have included robust quantification and lower magnification images. We used a Pearson's correlation coefficient (doi.org/10.1002/cyto.a.20896) to quantify our colocalization data and provide a statistical evaluation of the results.
- As for the other VAP27 proteins the work is in progress. Functional analyses of ER-plasma membrane contact sites have been done with the same two VAP27 proteins used in our study (e.g., two earlier studies: doi.org/10.1016/j.cub.2014.05.003; doi.org/10.1016/j.celrep.2018.04.091). Our conclusions are based on a functional characterization of the same VAP27 proteins at the ER-chloroplast contact sites. Providing only localization experiments in transient expression as suggested by the reviewer may not be conclusive without additional experimentation.

It is also worth noting that the expression pattern of these proteins when using their own promoters is very different from the transient expression experiments or the Arabidopsis lines using the 35S promoter.

- The VAP27 proteins fused to a fluorescent protein fusion are functional (doi.org/10.1016/j.cub.2014.05.003). As shown for the VAP27 proteins at the ER-plasma membrane contact sites, VAP27 proteins transiently expressed in protoplasts, tobacco leaf epidermis and stable transgenics are similarly distributed to these sites, as also compared with antibodies to the native proteins (doi.org/10.1016/j.cub.2014.05.003). However, we agree that transiently overexpressing the proteins, the signal of VAP27 proteins is more visible at the ER-plasma membrane contact sites compared to stable lines. Therefore, the differences appear to be quantitative (ER-plasma membrane) rather than qualitative (subcellular localization). As also shown previously for the localization of VAP27 at the ER-plasma membrane contact sites, in our work we have verified that the localization at ER subdomains is more marked as the expression levels of the protein increases. Indeed, in our work, the subcellular localization of VAP27 proteins driven by the native promoter (see Figure S1) is similar in subcellular localization to the VAP27 proteins expressed under the control of constitutive promoters, but signal is weaker.

Lines 156 to 157: Authors have analyzed VAP27-1 and VAP27-3 subcellular localization in Tobacco and Arabidopsis. They have found out that the stable expression with their own

promoters is different when they used constitutive promoters (lines 150-155). I do not think it has been proven that the subcellular localization of all VAP27 proteins is conserved interspecies.

- We would like to point the reviewer to the work of Wang et al., 2014 (tobacco and Arabidopsis; doi.org/10.1016/j.cub.2014.05.003), Liwen Jiang lab (doi.org/10.1073/pnas.2205314119) and our own – this and previous (doi.org/10.1016/j.celrep.2018.04.091). A comparison of these results in these manuscripts shows that the subcellular localization is conserved interspecies. Also please refer to “Plant VAP27 proteins: domain characterization, intracellular localization and role in plant development Pengwei Wang et al 2016 ” (doi.org/10.1111/nph.13857) in which *N. tabacum* or *N. benthamiana* were used for transient expression and then the protein localization was validated in the native system, in this case Arabidopsis. In the text we highlighted the protein distribution in the various backgrounds and indicated that VAP27 protein fusions expressed in stable Arabidopsis transgenics under the control of the respective endogenous promoters marked EMPC, which showed a more punctate appearance compared to EMPC visualized by VAP27 proteins expressed transiently or under the control of a constitutive promoter, most likely because of different levels of expression of the transgenes. Therefore, the results indicate that the differences are qualitative rather than quantitative, i.e., the VAP27 proteins mark the EMPC but to a different extent depending on the levels of expression. Wang et al. noted the same in their original work on ER-PM contact sites using VAP27 proteins (doi.org/10.1016/j.cub.2014.05.003).

A feature of BiFC is the irreversibility of the FP reconstitution. And it frequently stabilizes self-assemble FP partners at high local concentrations. Thus, a negative control is important to draw a clear conclusion in this experiment.

- As requested by the reviewer, negative controls have been added to the revised manuscript. Specifically, we used an ER membrane protein with similar orientation as VAP27, RHD3 (doi.org/10.1111/j.1365-313X.2011.04846.x). The negative control came completely clear demonstrating the accuracy of our experiment. This result has been added in a Supplemental Figure (Fig. S2A).

- Section 2: VAP27-1 and VAP27-2 interact with ORP2A...

Authors have proven the interaction of these proteins by Y2H and in vitro pulldown experiments. But they have already generated Arabidopsis stable lines expressing VAP27-CFP and ORP2A-YFP. I think they could provide stronger evidence for this interaction by performing Co-IP of these two proteins in Arabidopsis. Then, they will have evidence of the in vivo interaction. Additionally, to get better evidence of this interaction in vivo and at the PLAMS they could perform FRET-Flim experiments. A better negative control for the in vitro pulldown assays would also be good. For example, could you perform similar experiments with truncated version of these proteins? (It will also help in the identification of the interaction area). Or with any of the

other VAP27 proteins (for a negative interaction?) Do all VAP27 proteins interact with ORP2A? This is important to know when discussing sterol analysis (below).

- We thank the reviewer for this comment: A recent publication that was probably in the pipeline at the same time of our work shows that VAP27 interact with ORP2A at the ER-plasma membrane contact sites (doi.org/10.1073/pnas.2205314119), additionally supporting the interaction we found in our work. In our work, we provided two lines of evidence for an interaction of VAP27 with ORP2A and established that the interaction occurs at chloroplasts using yeast- two hybrid library screening and direct in vitro pull down. Therefore, additional experimentation should not be necessary. Also because of the interference of the chlorophylls, FRET-FLIM would not provide reliable results and likely provide a positive FRET signal as a result of an artifact.

Figure 4C: I would suggest the authors to show a bigger area of the cells. Are they only colocalizing at this subcellular area? Have they performed any quantification?

- This a very helpful suggestion. Because of the size of the EMPC, a larger area of the cells would most likely be confusing and not allow to see the sites. Additionally, the focal plane acquired with confocal microscopy is very defined and not all the chloroplasts are at the same focal plane acquired during the experiment. Therefore, to improve the presentation of our observations, we focused and acquired images to show the localization in the immediate surrounding areas of the chloroplasts, zoomed in to better show all the details, which would be hidden in a lower magnification. Moreover, in the revised manuscript we added Z-stack and maximum projection to illustrate a broader volume of the cell. We really appreciate the suggestion to do a quantification, which we did. The new data are included in Fig. S1A.

-Section 3: ORP2A and VAP27 proteins associate with the OEM

Authors have performed several experiments to gain insight into the association of these proteins with the OEM by using Trysin. These experiments have clearly demonstrated that the interaction of ORP2A with OEM. However, their studies about the binding of ORP2A and VAP27 to lipids is poor. I would suggest using a bigger collection of lipids (including PI, PC, PS, PE, DAG, TAG, sterols...)

- The interaction of VAP proteins and ORP2A with a larger collection of lipids has been already tested and published (doi.org/10.1016/j.celrep.2018.04.091; doi.org/10.1073/pnas.2205314119). We made sure to highlight this in the revised manuscript. In our work we purposely wanted to focus on specific lipids of the OEM to provide new information and test the interaction of VAP27 and ORP2 with those that had not been already reported. Therefore, our approach has expanded the lipid panels for VAP27 interactors.

Line 255: I do not think it has been proven these proteins form a complex.

- We established that the proteins form a complex on the bases of the interaction tested biochemically and by yeast two hybrid doing an Arabidopsis library screening, direct Y2H and an in vitro pull down. These results support other recent findings that VAP27 and ORP2A interact with each other (doi.org/10.1016/j.celrep.2018.04.091; doi.org/10.1073/pnas.2205314119). Additionally, the genetic evidence provided in this work that the VA27-ORP2A high order mutant has a similar phenotype to the ORP2A single mutant supports that the conclusions that they are involved in the same molecular complex.

Line 256: I do not think it is proven these proteins are binding “OEM signature glycerolipids”.

- We tested the direct binding of VAP27-1, VAP27-3 and ORP2A with MGDG, DGDG and PG. These lipids are present in the outer envelope membrane, and they are specific for plastids (doi.org/10.1146/annurev-arplant-050718-100202) hence we called them OEM signature glycerolipids. We rephrased this sentence to avoid the term.

- Section 4: The loss of VAP27 and ORP2A...

I agree with the authors, changes in glycerolipids are very subtle.

- The glycolipids we were mainly interested in this work were MGDG, DGDG, SQDG and PG, which are the most abundant lipids in photosynthetic membranes. Indeed, the biosynthesis of galactolipids is regulated in response to chloroplast functionality. Therefore, analyzing the content of these lipids was the first step toward understanding the role of VAP27 and ORP2A in lipid transport at the ER-chloroplast interface. The differences we identified are subtle but still significant, and informative.

- Section 5: ORP2A is necessary for sterol homeostasis

Authors have analyzed major sterols in WT, *orp2a-1*, *vap27-1/vap27-3*, and *vap27-1/vap27-3/orp2a-1* mutants. And they interestingly have found out that mutant plants accumulate more sterols (in particular sitosterol and campesterol) than WT plants. However, it is surprising that *vap27-1/vap27-3* is accumulating sterols at the same levels as the *orp2a-1* and the triple mutant. These could point to *vap27-1* and *-3* being the two only VAP27 proteins at the PLAMS. I would suggest to better discuss this in the discussion.

- We thank the reviewer for this observation. Indeed, the fact that *vap27-1/vap27-3* is accumulating sterols at the same levels as the *orp2a-1* and the triple mutant is genetic evidence that VAP27-1 VAP27-3 and ORP2A are likely part of the same machinery involved in the process of sterol transport from ER to chloroplasts. If they had not worked together, we would have found an additive effect on sterols accumulation. We edited the discussion in the revised manuscript to underline this concept.

I also think it could be interesting to show a picture of these plants (in the supplementary part). Are they showing the same phenotype?

- We agree with the reviewer that picture a complete set of the mutants would be interesting to be shown. We added the requested data in Figure S3.

Finally, as chloroplast are not the main site for sterol accumulation, it would be good if the authors can show the sterol composition in all total tissue like leaves (not just in chloroplast). Did they notice any changes?

- The goal of this experiment is to quantify the levels of the most representative sterols that have been reported so far at chloroplasts (doi.org/10.1104/pp.47.6.745) to discriminate any possible difference in a putative role of VAP27-1/ -3 and ORP22A in transport from the ER to the chloroplasts comparing the WT with the different mutants. The amount of sterols at chloroplast level is relatively low and we purposely preferred, planned and proceeded an accurate purification of chloroplasts vs a total sterols leaf extraction in order to be able to appreciate small difference in the level of sterols at chloroplast level. By doing a total extract, if any difference due to a defect in an activity of transport between ER and chloroplast were present, it would have been masked by the larger amount of sterols that would have saturated our quantification, being the plasma membrane the site of highest accumulation of sterols and the ER the organelle that initiates sterol biosynthesis. We explained the reason for this experimental approach better in the results and we thank the reviewer for bringing up this issue.

Discussion:

Line 338: I do not think it is completely right to assert that VAP27-1 and VAP27-3 proteins preferentially accumulate at PLAMs.

- We have edited the text to read: “The plant ER connects with chloroplasts at regions where VAP27 proteins exhibit reduced mobility and interact with the OEM”.

Lines 336 and 337: I do not think it is proven. I think you should add other lipids to the analysis so they can be negative controls.

- This point refers to our section in the discussion where from line 336 to 339 where we state: “In this work, we provide evidence that two ER membrane associated VAP27 proteins, VAP27-1 and VAP27-3, interact with ORP2A at ER subregions that interface with chloroplasts, where they preferentially accumulate and physically associate with the OEM”. The reason why in line 338 we assert that VAP27-1 and VAP27-3 proteins preferably accumulate at PLAMs which stands for “Plastid Associated Membranes” is because we have both live cell microscopy where we can see that the levels of the two proteins are higher in the surrounding area of the chloroplast compared to the ER tubules. This is supported by additional data presented in the manuscript, namely: 1) confocal microscopy evidence by BiFC that the VAP27 proteins are in close proximity with the chloroplast outer envelope, 2) analysis of protein recovery after photobleaching

demonstrating that the mobility of VAP27 proteins at the ER interfacing with chloroplast is lower compared to the rest of the ER, and 3) biochemical evidence that the VAP27 proteins can interact with the outer envelope. Therefore, with these results and previous microscopy and biochemical evidence that the ER and chloroplasts are connected physically and biochemically, we believe that we can go ahead in supporting the hypothesis that VAP27-1 and VAP27-3 proteins preferentially accumulate at PLAMs. Several lipids have been tested for interaction with VAP27 proteins in earlier work (e.g., doi.org/10.1016/j.celrep.2018.04.091; doi.org/10.1073/pnas.2205314119). In this work we focused and demonstrated that there is a direct interaction of VAP27 proteins with signature lipids of the outer envelope, which had not been tested in previous publications.

Lines 401-403: It is really difficult to understand how VAP27 proteins, that are known to be localized at other MCSs, are binding MGDG. I think authors should discuss this.

- As the reviewer suggested, this has been better discussed.

Reviewers' Comments:

Reviewer #1:

Remarks to the Author:

Renna et al. report that VAP27 forms a functional complex with ORP2A, thus bridges the contact sites between OEM and ER which are involved in chloroplast lipid homeostasis. This work investigates the specific membrane contact sites between ER and chloroplast which are important and unique in plants. Therefore, it is helpful to expanding our understanding of the function of membrane contact sites.

The study is well organized and the revised manuscript has addressed most of the questions I concerned. However, I think that the model in Fig8 is rather simple to cover the whole story. Moreover, the description in the figure legends beyond the information in the figure.

Reviewer #2:

Remarks to the Author:

The authors answered correctly to most of my concerns and I really thinks their work raises an intriguing question on the role of chloroplast in sterol homeostasis. However, to really apprehend this part, I think a whole sterol quantification at the leaf level is important to know if this accumulation of sterol is linked to a change in sterol synthesis. Furthermore, to have an idea of how big the sterol chloroplast pool is, in comparison with the total cell sterol pool, is necessary. To achieve that comparison, maybe the normalization of sterol content per nmol of fatty acid or per mg of protein could be used. Both analysis are important at the chloroplast level and at the cell level, especially because the double mutant vap27 does not have the same growth phenotype than the ORP2A mutant and that might be link also to a defect in global sterol homeostasis.

In the same way, the glycerolipid analysis are presented only in mol % whereas there is a growth phenotype in the mutants. Is there a global defect in lipid synthesis? At least, the total fatty acid content per mg of fresh weight or dry weight should be shown because if there are more grana in the mutants, there are maybe more lipids as well...

Minor remark:

- Fig S4 Legend is not clear: I guess chlorophyll autofluorescence is in blue, DiOC6 in purple. What are the arrows representing?

- In the final scheme, only MGDG and PG are presented to link ORP2A to chloroplast. However, it was shown that ORP2A binds phosphoinositides. What about their presence in chloroplast?

Reviewer #3:

Remarks to the Author:

The authors have submitted a revised version of their manuscript and they have largely answered many of my questions/comments and supplied additional information, consequently, the manuscript's quality has noticeably increased. They have added controls to the BiFC experiment and showed Z-Stacks maximum projection images, convincingly proving that VAP27-1 and VAP27-3 interacts with chloroplast. Overall, the revision seems to address most of the concerns raised by the reviewers. As such, I find the manuscript suitable for publication in Nature Communications, but there are several weaknesses that I think they should address prior to acceptance for publication. In no order, these include:

(1) I suggested to specify VAP27-1 and VAP27-3 in the title, and authors agreed to do so.

However, I have not seen this change in the revised version.

(2) Lines 71 and 72 – do not seem to be right. Please check them.

(3) Figure 5B. Although, the interaction of VAP and ORP2A proteins have already been proven in other publications (as authors remarked in the review process), this work should include some of

other phospholipids in the protein-lipid overlay assays (as positive/negative controls). Problems with promiscuous binding occur frequently in these assays, thus, to clearly prove protein-lipid interactions, proper controls need to be added. Results are not that clear, and authors are strongly claiming the binding to MGDG and PG.

(4) Figure 7E. I have the same concerns as in Figure 5B, many controls are missing. Authors could just add other sterols (cholesterol?) to check if there is a specific binding of these 3 plant sterols or not. Negative controls are strongly needed. They could add DGDG (as they have previously shown ORP2A is not binding them).

Authors' comments:

We would like to thank the reviewers for their insightful comments. We have addressed all the comments by editing the manuscript text and incorporating the requested experiments. We are grateful for the reviewers' input because addressing their suggestions has strengthened our work. Detailed responses to the reviewers follow below.

Reviewer #1 (Remarks to the Author):

Renna et al. report that VAP27 forms a functional complex with ORP2A, thus bridges the contact sites between OEM and ER which are involved in chloroplast lipid homeostasis. This work investigates the specific membrane contact sites between ER and chloroplast which are important and unique in plants. Therefore, it is helpful to expanding our understanding of the function of membrane contact sites.

The study is well organized and the revised manuscript has addressed most of the questions I concerned. However, I think that the model in Fig8 is rather simple to cover the whole story. Moreover, the description in the figure legends beyond the information in the figure.

Author response: Thank you for your positive assessment. We see the reviewer's point and we have provided a more complete model and figure legend.

Reviewer #2 (Remarks to the Author):

The authors answered correctly to most of my concerns and I really thinks their work raises an intriguing question on the role of chloroplast in sterol homeostasis. However, to really apprehend this part, I think a whole sterol quantification at the leaf level is important to know if this accumulation of sterol is linked to a change in sterol synthesis. Furthermore, to have an idea of how big the sterol chloroplast pool is, in comparison with the total cell sterol pool, is necessary. To achieve that comparison, maybe the normalization of sterol content per nmol of fatty acid or per mg of protein could be used. Both analysis are important at the chloroplast level and at the cell level, especially because the double mutant *vap27* does not have the same growth phenotype than the *ORP2A* mutant and that might be link also to a defect in global sterol homeostasis. In the same way, the glycerolipid analysis are presented only in mol % whereas there is a growth phenotype in the mutants. Is there a global defect in lipid synthesis? At least, the total fatty acid content per mg of fresh weight or dry weight should be shown because if there are more grana in the mutants, there are maybe more lipids as well...

Author response: We appreciate the point of the reviewer that differences in sterol content in chloroplasts might be linked to availability of total sterols of the various genetic backgrounds analyzed in our work. Therefore, to test if the increased accumulation of β -sterol and campesterol observed in the mutants (Fig. 7D) could be linked to a change in total sterol availability, we proceeded with a sterol total leaf extraction and quantification of the sterols previously quantified in chloroplasts only. We normalized the data to protein content, which was found to be indistinguishable across genotypes (Figure S8B). Of note, chlorophyll content also did not yield significant differences across genotypes (Figure S8C). We analyzed all genotypes (i.e., WT, *orp2a-1*, *vap27-1/vap27-3*, and *vap27-1/vap27-3/orp2a-1*) and found no statistically significant differences across them, except for a slight reduction (6%) in the levels of β -sitosterol in the triple *vap27-1/vap27-3/orp2a-1* mutant (Figure S8A). Because the levels of β -sterol and campesterol in chloroplasts are higher in all the mutants compared to WT (Figure 7D), the absence of differences in the total leaf sterol content between WT and *orp2a-1* and *vap27-1/vap27-3* and the slight reduction of total β -sitosterol in the *vap27-1/vap27-3/orp2a-1* mutant is unlikely causative of the observed increase of sterol levels in chloroplasts. We thank the reviewer for this comment as it helped us consolidate our initial observations. Concerning the glycerolipids analyses, the reason we determine mol % ratios for membrane lipids was to test if the mutant lines are affected in lipid transfer from the ER through the contacts with chloroplasts. Past experience from our labs and others with genuine ER to plastid lipid transfer mutants (e.g., doi/pdf/10.1093/emboj/cdg234; doi.org/10.3390/ijms21155325) indicates that the ratio of 16:3 to 18:3 fatty acids particularly in MGDG is diagnostic because of differences in acetyltransferase specificity for ER and plastid derived precursors for MGDG biosynthesis. As presented in Figure 6 of the manuscript, we did not see a difference in the ratio of 16:3 to 18:3 fatty acids and our conclusion was that ER to plastid lipid trafficking was not affected in this mutant. Trying to determine what causes the small growth phenotype of the mutant lines is a different question. The reviewer suggests that this could be due to a decrease in lipid biosynthesis, which is a valid point, but it could be equally due to other factors such as a decrease in specific amino acid biosynthesis, over production of the plant hormone JA, which has been reported for some lipid mutants or countless other reasons. Therefore, investigating the true cause of the growth phenotype would be a major undertaking beyond the scope of this paper. To more specifically answer the reviewer's question regarding a possible defect in lipid biosynthesis, we note that membrane lipid biosynthesis which dominates the fatty acid pool in green tissues typically tracks with chlorophyll content as a reflection of the extent of thylakoid membranes. In the revised version of the manuscript, we have added chlorophyll (Figure S8C) and total protein measurements (Fig. S8B) showing that there are no significant differences across genotypes. As we show that chlorophyll content is not altered in the mutants, we therefore can conclude that lipid biosynthesis is likely not altered either. Based on these

arguments and the new added data on chlorophyll quantification, we feel that the suggested experiment, which would take time to regrow plants and analyze them, does not provide additional information to answer questions we are trying to address here.

Minor remark:

- Fig S4 Legend is not clear: I guess chlorophyll autofluorescence is in blue, DiOC6 in purple. What are the arrows representing?

Author response: We amended Figure S4 legend to make it more readily accessible as requested by the reviewer.

- In the final scheme, only MGDG and PG are presented to link ORP2A to chloroplast. However, it was shown that ORP2A binds phosphoinositides. What about their presence in chloroplast?

Author response: There are works reporting the presence of phosphoinositides in the chloroplasts (<https://doi.org/10.1016/j.bbabc.2013.09.007>) and there is evidence reporting that ORP2A binds phosphoinositides (<https://doi.org/10.1093/plphys/kiad238>). The reason why we did not take them in consideration in a first place is because their localization is not specific to chloroplasts, therefore we focused on lipids (glycolipids) that are exclusive of the chloroplast to determine the interaction of the proteins with them. We have amended the figure legend of the final scheme indicating that ORP2A might also bind to phosphoinositides in the chloroplast outer membrane membranes. We had addressed this point in the discussion in the original submission and in light of the reviewer's comment, we have kept it in this submission.

Reviewer #3 (Remarks to the Author):

The authors have submitted a revised version of their manuscript and they have largely answered many of my questions/comments and supplied additional information, consequently, the manuscript's quality has noticeably increased. They have added controls to the BiFC experiment and showed Z-Stacks maximum projection images, convincingly proving that VAP271 and VAP27-3 interacts with chloroplast. Overall, the revision seems to address most of the concerns raised by the reviewers. As such, I find the manuscript suitable for publication in Nature Communications, but there are several weaknesses that I think they should address prior to acceptance for publication. In no order, these include:

(1) I suggested to specify VAP27-1 and VAP27-3 in the title, and authors agreed to do so. However, I have not seen this change in the revised version.

Author response: We apologize for the oversight. We have amended the title and specified VAP27-1 and VAP27-3 as the reviewer suggested.

(2) Lines 71 and 72 – do not seem to be right. Please check them.

Author response: Thank you for pointing our attention to those lines. We amended the text accordingly.

(3) Figure 5B. Although, the interaction of VAP and ORP2A proteins have already been proven in other publications (as authors remarked in the review process), this work should include some of other phospholipids in the protein-lipid overlay assays (as positive/negative controls). Problems with promiscuous binding occur frequently in these assays, thus, to clearly prove protein-lipid interactions, proper controls need to be added. Results are not that clear, and authors are strongly claiming the binding to MGDG and PG.

Author response: We see the point of the reviewer. Accordingly, we performed the experiment again using very stringent conditions, and positive and negative controls based on published literature. Using these stringent conditions, we confirmed the interaction with positive controls and a lack of interaction with the negative controls, supporting the specificity of our assays. Repeating the experiments with more stringent conditions allowed us to obtain very clean results, and we could confirm the interaction only with MGDG. Using these stringent conditions we did not verify an interaction with PG. It is possible that the interaction with PG is weak. We therefore thank the reviewer for raising this important point and helping us to improve the data quality. We have amended the Materials and Methods section with the revised procedures used for the data presented in the manuscript.

(4) Figure 7E. I have the same concerns as in Figure 5B, many controls are missing. Authors could just add other sterols (cholesterol?) to check if there is a specific binding of these 3 plant sterols or not. Negative controls are strongly needed. They could add DGDG (as they have previously shown ORP2A is not binding them).

Author response: As requested by the reviewer, we added DGDG as a negative control. We also added cholesterol, which does not interact with ORP2A in our assay, supporting the specificity of the interaction of ORP2A with the other sterols used in the analysis.

Reviewers' Comments:

Reviewer #2:

Remarks to the Author:

The authors answered correctly to my concerns. With the new data provided on the sterols, I'm convinced by their statements.

As such, I find the manuscript suitable for publication in Nature Communications.

Reviewer #3:

Remarks to the Author:

I am satisfied with the revision, authors addressed my concerns.